# Parallel processing by distinct classes of principal neurons in the olfactory cortex

Shivathmihai Nagappan, Kevin M Franks*

Department of Neurobiology, Duke University Medical School, Durham, United States

**Abstract** Understanding how distinct neuron types in a neural circuit process and propagate information is essential for understanding what the circuit does and how it does it. The olfactory (piriform, PCx) cortex contains two main types of principal neurons, semilunar (SL) and superficial pyramidal (PYR) cells. SLs and PYRs have distinct morphologies, local connectivity, biophysical properties, and downstream projection targets. Odor processing in PCx is thought to occur in two sequential stages. First, SLs receive and integrate olfactory bulb input and then PYRs receive, transform, and transmit SL input. To test this model, we recorded from populations of optogenetically identified SLs and PYRs in awake, head-fixed mice. Notably, silencing SLs did not alter PYR odor responses, and SLs and PYRs exhibited differences in odor tuning properties and response discriminability that were consistent with their distinct embeddings within a sensory-associative cortex. Our results therefore suggest that SLs and PYRs form parallel channels for differentially processing odor information in and through PCx.

## Editor's evaluation

In this timely study, Nagappan and Franks challenge a long-held model of sequential odor processing by two classes of excitatory principal neurons in the olfactory cortex (semilunar neurons and superficial pyramidal neurons). The main findings of this study support an alternate interpretation that semilunar and pyramidal neurons process olfactory information in parallel rather than sequentially. This work is highly relevant to the olfactory field as well as those scientists who aim to understand the roles of diverse neural classes in sensory processing. The work is executed very well, properly controlled and analysed, and the claims are very well supported by the data.

*For correspondence:
franks@neuro.duke.edu

**Competing interest:** The authors declare that no competing interests exist.

## Introduction

The brain contains a diverse array of neuron types that are each thought to play a specific role in neural circuit computations (*Sanes and Masland, 2015*; *Kim et al., 2020*; *Yao et al., 2021*). Understanding the specific role each neuron type plays will therefore provide mechanistic insight into what a neural circuit does and how it does it (*Zeng and Sanes, 2017*). Piriform cortex (PCx) is a three-layered allocortex with principal neurons primarily located in layer II. Superficial pyramidal (PYR) cells in deeper layer II receive afferent olfactory bulb (OB) inputs on their distal apical dendrites, dense excitatory intracortical inputs on their proximal apical dendrites, and top-down inputs from other brain regions on their basal dendrites. By contrast, semilunar (SL) cells in superficial layer II receive strong OB inputs on their distal apical dendrites, few, if any, excitatory intracortical inputs, and lack basal dendrites (*Figure 1a*; *Heimer and Kalil, 1978*; *Haberly, 1983*; *Bekkers and Suzuki, 2013*). These differences in input sources and local connectivity between SLs and PYRs have motivated predictions about how odor information is routed through and processed within PCx.

In the prevailing model, odor information flows sequentially through SLs and PYRs and is processed hierarchically, first by SLs and then by PYRs. Support for this sequential model stems from observations that SLs receive stronger OB inputs than PYRs but little or no intracortical excitatory inputs, and that SLs respond 2–4 ms earlier than PYRs following electrical stimulation of OB mitral cell axons in the lateral olfactory tract (LOT) (*Ketchum and Haberly, 1993*; *Suzuki and Bekkers, 2006*; *Suzuki and Bekkers, 2011*; *Wiegand et al., 2011*; *Choy et al., 2017*). According to this model, SLs are thought to be the primary recipients of OB input, and their activity necessary to recruit PYRs. Alternatively, SLs and PYRs may form parallel processing streams, such as described in various sensory and motor neocortices, where principal neurons projecting to distinct downstream regions encode different features of a stimulus or are differentially modulated during task learning (*Glickfeld et al., 2013*; *Chen et al., 2015*; *Economo et al., 2018*; *Williamson and Polley, 2019*). That SLs and PYRs project to distinct downstream regions (*Diodato et al., 2016*; *Mazo et al., 2017*) lends support to a parallel model. Finally, SLs and PYRs may be embedded within a parallel loop where mitral cells project onto both SLs and PYRs, and SLs additionally project onto PYRs, similar in organization to the entorhinal cortex-dentate gyrus-CA3 circuit in the hippocampus (*Hainmueller and Bartos, 2020*).

PCx is a sensory-associative cortex, containing both afferent OB inputs and recurrent or intracortical excitatory inputs (*Johnson et al., 2000*; *Franks et al., 2011*; *Suzuki and Bekkers, 2011*; *Wiegand et al., 2011*; *Hagiwara et al., 2012*). The dual afferent and associative inputs provide PCx neurons the substrate to integrate elemental sensory information from the OB to form distributed odor-specific neuronal ensembles and transform these ensembles through the recurrent, associative network (*Barkai et al., 1994*; *Haberly, 2001*; *Poo and Isaacson, 2011*; *Bolding and Franks, 2018*; *Bolding et al., 2020*; *Pashkovski et al., 2020*). SLs and PYRs could represent the two ends of the PCx sensory-associative continuum, where SLs simply combine afferent OB inputs while PYRs form dynamic odor ensembles that can be transformed through the recurrent, associative network. If so, we might expect SL odor representations to reflect sensory information more faithfully and PYR odor representations to benefit from associative learning mechanisms and be more reliable and discriminable.

We recorded extracellularly from layer II PCx neurons in awake, head-fixed *NetrinG1Cre* (*Ntng1Cre*) mice that selectively express Cre-recombinase in SLs (*Bolding et al., 2020*). This allowed us to distinguish SLs from PYRs in populations of layer II PCx neurons and to selectively manipulate SL activity. We found that neither optogenetic nor chemogenetic suppression of SLs affected PYR odor responses and that SLs and PYRs responded with similar latencies, arguing against the sequential model. SLs were more broadly tuned, and odors were more discriminable in PYRs than in SLs, suggesting that distinct principal neuron types in PCx differentially encode odors, and in ways that reflect their local connectivity profiles. Together, these findings support a parallel processing model, where SLs and PYRs both receive direct OB inputs and transform the information to varying degrees, perhaps for use in different behavioral contexts by their distinct downstream targets.

## Results

### *Ntng1*Cre transgenic mouse enables selective targeting of SLs in vivo

We recently generated a knock-in mouse line that expresses Cre-recombinase in Ntng1-expressing (Ntng1+) cells throughout the brain (*Bolding et al., 2020*; *Figure 1—figure supplement 1*). In PCx, Cre-expressing cells were largely restricted to the superficial half of layer II (i.e., layer IIa), where SLs are located (*Figure 1b*). Ntng1+ cells displayed hallmark morphological characteristics of SLs: half-moon-shaped somata, two prominent apical dendrites emerging directly from the cell body and no basal dendrites (percent Ntng1+ cells with SL morphology, mean ± SD: 87.9% ± 3.54%, n = 3 mice; *Figure 1—figure supplement 2*). Additionally, Ntng1+ and Ntng1- cells exhibited the intrinsic biophysical properties described in SLs and PYRs, respectively (*Figure 1c–h*; *Suzuki and Bekkers, 2006*; *Suzuki and Bekkers, 2011*). Previous studies have reported that SLs form excitatory synaptic connections onto PYRs and local inhibitory interneurons but receive few intracortical excitatory inputs (*Suzuki and Bekkers, 2011*; *Choy et al., 2017*). To confirm that this synaptic organization was also true for Ntng1+ and Ntng1- cells, we injected Cre-dependent adeno-associated viruses (AAVs) into the PCx of *Ntng1Cre*/Ai14 mice to express channelrhodopsin-2 (ChR2) in a focal subset of Ntng1+ cells (*Figure 1—figure supplement 3a*). We then obtained whole-cell voltage-clamp recordings from uninfected Ntng1+ and Ntng1- cells located away from the infection site (*Figure 1—figure supplement*

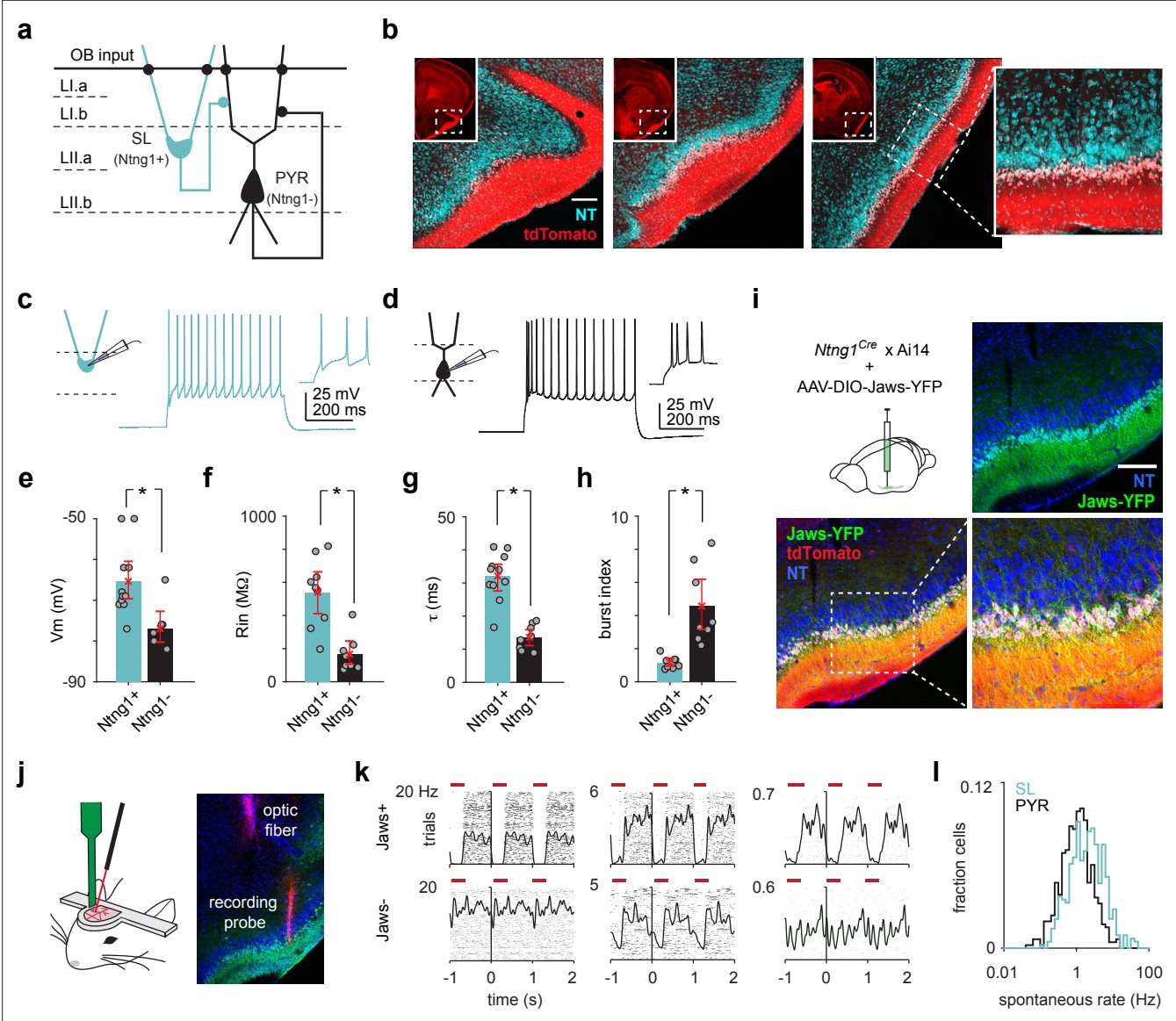

**Figure 1.** Semilunars (SLs) can be optogenetically identified from extracellular recordings of awake, head-fixed *Ntng1^Cre* mice. (**a**) Schematic of SL and pyramidal (PYR) connectivity and input sources within piriform cortex (PCx). (**b**) Coronal sections from a *Ntng1^Cre*/Ai14 mouse brain showing tdTomato labeling from anterior to posterior PCx. tdTomato labeling is restricted to superficial layer II of PCx. Scale bar is 200 μm. (**c**) Whole-cell current-clamp recordings were obtained from acute brain slices isolated from *Ntng1^Cre*/Ai14 mice. Example voltage responses to direct current injections for a Ntng1+ (tdTomato+) cell. (**d**) As in (**a**). but for a Ntng1- (unlabeled) cell. (**e**) Resting membrane potentials of Ntng1+ and Ntng1- cells (Ntng1+: –65.4 mV [–70,–60], n = 11 cells; Ntng1-: –77.0 mV [–80.6,–72.9], n = 7 cells; p=5.27 × 10^{-4}, unpaired t-test). (**f**) Input resistances of Ntng1+ and Ntng1- cells (Ntng1+: 541 MΩ [415, 669], n = 9 cells; Ntng1-: 166 MΩ [104, 243], n = 8 cells; p=2.15 × 10^{-4}, unpaired t-test). (**g**) Membrane time constants of Ntng1+ and Ntng1- cells (Ntng1+: 32.1 ms [27.7, 35.8], n = 11 cells; Ntng1-: 13.6 ms [11.2, 16.0], n = 8 cells; p=1.29 × 10^{-5}, unpaired t-test). (**h**) Burst indices of Ntng1+ and Ntng1- cells (Ntng1+: 1.16 [0.926, 1.41], n = 8 cells; Ntng1-: 4.57 [3.12, 6.19], n = 8 cells; p=1.43 × 10^{-4}, unpaired t-test). (**i**) Coronal section of anterior PCx from an experimental *Ntng1^Cre*/Ai14 mouse showing selective and robust expression of the inhibitory opsin, Jaws, in SLs. Scale bar is 200 μm. (**j**) Schematic of recording probe and optic fiber positioning for opto-tagging (left). Histology showing actual positioning of the recording probe and optic fiber in PCx after both were painted with DiI (right). (**k**) Raster plots with trials aligned to the onset of each light pulse for six example units that were later categorized as either Jaws+ (top) or Jaws- (bottom). (**l**) Distribution of spontaneous spike rates for SLs (blue, n = 426 cells) and PYRs (black, n = 464 cells).

The online version of this article includes the following figure supplement(s) for figure 1:

**Figure supplement 1.** Ntng1+/Cre+ neurons throughout the brain.

**Figure supplement 2.** Ntng1+ cells have similar laminar and morphological properties as semilunars (SLs).

**Figure supplement 3.** Ntng1+ and Ntng1- cells have similar synaptic connectivity properties as semilunars (SLs) and pyramidals (PYRs).

*Figure 1 continued on next page*

*Figure 1 continued*

**Figure supplement 4.** Jaws expression is consistent across mice.

**Figure supplement 5.** Semilunars (SLs) are identified using inhibitory opto-tagging.

**Figure supplement 6.** Waveform analyses do not unambiguously distinguish between excitatory and inhibitory neurons.

*3c*). We observed large, rapid inward currents in all recorded Ntng1- cells in response to brief (1 ms) light pulses but almost no inward currents in Ntng1+ cells (*Figure 1—figure supplement 3d and e*). These light-evoked responses were completely blocked by glutamate receptor antagonists (NBQX and CPP) (*Figure 1—figure supplement 3b*). Thus, Ntng1+ cells make glutamatergic synapses onto Ntng1- cells, but not onto other Ntng1+ cells, consistent with the synaptic connectivity of SLs and PYRs. Taken together, the *Ntng1$^{Cre}$* mouse expresses Cre-recombinase in a subpopulation of PCx principal neurons whose laminar, morphological, intrinsic, and local connectivity properties are congruent with those of SLs. Therefore, henceforth we refer to Ntng1+ and Ntng1- cells within layer II of PCx as SLs and PYRs, respectively.

Although SLs do not receive excitatory inputs from other SLs, it is not known if they receive excitatory inputs from other, non-SL PCx neurons. We therefore obtained whole-cell voltage-clamp recordings in mice focally injected with non-conditional AAVs to express ChR2 in all PCx neurons (*Figure 1—figure supplement 3f*). We observed even larger rapid inward currents in PYRs, consistent with the dense recurrent connectivity between PYRs. Interestingly, we now observed small inward currents in SLs, indicating that SLs do receive some intracortical excitatory inputs from other PCx neurons, although these were 5.4× weaker than inputs received by PYRs (*Figure 1—figure supplement 3g and h*).

To distinguish between SLs and PYRs in vivo, we first injected Cre-dependent AAVs expressing the inhibitory opsin, Jaws (*Chuong et al., 2014*), into anterior PCx of *Ntng1$^{Cre}$*/Ai14 mice (*Figure 1i*, *Figure 1—figure supplement 4b*). After allowing 3–4 weeks for Jaws expression, we recorded the extracellular spiking activity of layer II PCx neurons at the injected site in awake, head-fixed mice (*Figure 1j*). We used an optogenetic tagging strategy to classify recorded cells as SLs. We presented a series of brief 635 nm light pulses (1000 pulses, 300 ms, 1 Hz) to suppress spiking in Ntng1+ cells. Cells were classified as SLs if their activity was reliably (i.e., spike rate greater during baseline than during the light-on period, two-tailed t-test, $p < 1 \times 10^{-7}$) and rapidly (i.e., suppression latency determined using change point analysis <10 ms) suppressed by light (*Wolff et al., 2014*; *Rowland et al., 2018*; *Figure 1j and k*, *Figure 1—figure supplement 5*). Across all experiments, approximately half of the recorded cells were SLs (*Figure 1—figure supplement 4a*). To minimize false negatives due to poor infection and/or mistargeted recordings, we omitted experiments in which less than 25% of cells were suppressed. We previously found that only 7% of all recorded PCx neurons, and fewer than 1% of layer II PCx neurons, were GABAergic inhibitory neurons (*Bolding and Franks, 2018*). The remaining units were therefore classified as PYRs. SLs had higher spontaneous spike rates than PYRs (median SL: 1.99 Hz, n = 426 cells; median PYR: 1.03 Hz, n = 464 cells; $p = 5.08 \times 10^{-13}$, unpaired t-test; *Figure 1l*), indicating that we can resolve differences in spiking activity between SLs and PYRs. Importantly, our ability to tag neurons with very low spontaneous firing rates (e.g., *Figure 1k*, *Figure 1—figure supplement 5d*) indicate that our optogenetic tagging protocol was robust for identifying cells with low spontaneous firing rates. We did not find systematic differences in spike waveforms between SLs and PYRs (*Figure 1—figure supplement 6*).

## Optogenetic suppression of SLs does not weaken or reshape PYR odor responses

We first asked if odor information from the OB is processed sequentially in PCx, in which case PYRs would require SL activity to respond. We optogenetically suppressed SLs while recording baseline and odor-evoked activity in PYRs (*Figure 2a*). To maximize the area of PCx suppressed, we injected Cre-dependent Jaws in three-four sites along its anterior-posterior axis (*Figure 2b*, *Figure 2—figure supplement 1b*) and used a 400 µm optic fiber, positioned at an angle to the recording probe to deliver light (*Figures 1j and 2c*). We opto-tagged SLs at the end of each experiment (fraction of identified SLs, mean ± SD: 0.458 ± 0.0996, n = 9 experiments, six mice, *Figure 2—figure supplement 1a*). Every 10 s, we delivered 1-s-long odor pulses, triggered on exhalation (*Figure 2d*). On alternating

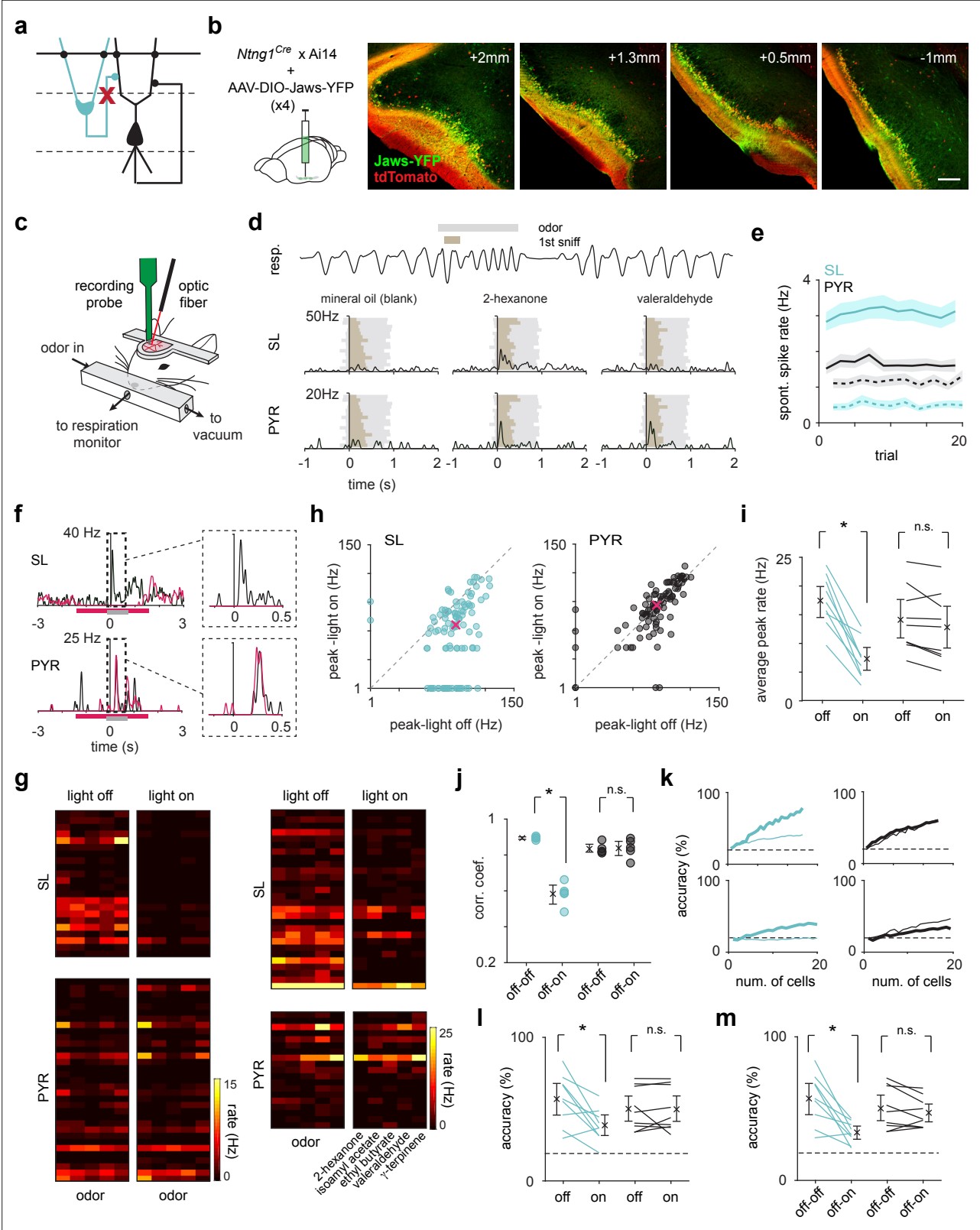

**Figure 2.** Optogenetic suppression of SLs does not weaken or reshape PYR odor responses. (**a**) SLs were selectively optogenetically suppressed during recording to reveal the role of SL input on PYRs. (**b**) The inhibitory opsin, Jaws, was injected in three-four sites along the anterior-posterior axis (left to right) of PCx to maximize SL suppression throughout PCx. Scale bar is 200 μm. (**c**) Schematic of experimental setup for recording odor responses from awake, head-fixed *Ntng1^Cre* mice. (**d**) Example respiration trace from one odor trial (top). A 1-s-long odor pulse (gray) is triggered on exhalation.

*Figure 2 continued on next page*

*Figure 2 continued*

Spiking activity was aligned to the onset of the inhalation of the first subsequent sniff (brown). Example trial-averaged SL and PYR responses (PSTHs) to the control stimulus and two different odors. (**e**) Average spontaneous spike rate of SLs (blue) and PYRs (black) during *light off* (solid line) and *light on* (dotted line) trials. (**f**) Example odor response from one SL and one PYR cell during *light-on* (magenta trace) and *light-off* (black trace) trials. The magenta bar indicates when the light was on, and the gray bar indicates when the odor was presented. (**g**) Heatmaps of trial-averaged spike rates in the first sniff for SL and PYR cell-odor pairs from two example experiments, one with complete (left) and one with incomplete (right) SL suppression. *Light-on* and *light-off* trials are plotted separately. (**h**) Odor-evoked response peaks for significantly activated SLs (blue, n = 101 cell-odor pairs) and PYRs (black, n = 97 cell-odor pairs) during *light-on* trials plotted against *light-off* trials. Median response peak is shown in magenta. (**i**) Mean odor-evoked response peaks within each experiment for each condition for all activated SLs and PYRs. Each point is an experiment (n = 9). (**j**) Pearson's correlation coefficients of response vectors between the *light-on* and *light-off* conditions (off-on, n = 5 odors) or in just the *light-off* condition (off-off, n = 5 odors) for each odor. Firing rates in the first sniff following odor presentation for individual neurons were averaged within the two trial sets (*light-off* and *light-on*). Then, the pseudopopulation response vector consisting of trial-averaged *light-off* responses was correlated with the pseudopopulation response vector consisting of trial-averaged *light-on* responses. (**k**) Odor classification accuracy determined using a linear support vector machine for SLs (blue) and PYRs (black) for two example experiments as a function of cell number in the *light-on* (thick line) and *light-off* (thin line) conditions separately. The decoder was asked to classify responses (spike counts in a 500 ms time window beginning at odor inhalation) on a single trial to the panel of five odors. (**l**) Summary plot for within condition decoding (as in **k**) for a size-matched population of SLs and PYRs for individual experiments. Each point is an experiment (n = 9). (**m**) Odor classification accuracy for SLs and PYRs in each experiment, where the classifier was trained on trials from the *light-off* condition and tested on trials from the *light-on* condition. Each point is an experiment (n = 9).

The online version of this article includes the following figure supplement(s) for figure 2:

**Figure supplement 1.** Spread and efficacy of Jaws expression in optogenetic suppression experiments.

odor trials, we delivered a 3-s-long light pulse, spanning 1 s before to 1 s after odor presentation to suppress SLs. Spiking activity was aligned to the inhalation phase of the first sniff following odor presentation (*Figure 2d*; *Bolding and Franks, 2017*).

During a 500 ms period before odor onset, we observed strong and sustained light-evoked suppression of SLs, as well as a slight decrease in PYR spike rates, indicating that SLs contribute to spontaneous PYR activity (*Figure 2e*). We then examined the odor-evoked spike rates of SLs and PYRs. To avoid confounds of odor-evoked suppression, we first only considered 'activated' responses during the *light-off* trials (response index > 0, see Materials and methods). During *light-on* trials, odor-evoked activity, measured as response peaks in a 300 ms period after odor inhalation, was markedly suppressed in SLs but remained unchanged in PYRs (*Figure 2f and g*). This was the case when we only considered robustly activated responses (median peak firing rate for significantly activated responses [p<0.05, Wilcoxon rank-sum test]: SL, *light-off*: 29.7 Hz, *light-on*: 13.0 Hz; n = 101 cell-odor pairs; p=2.71 × 10$^{-12}$, paired t-test; PYR, *light-off*: 24.9 Hz, *light-on*: 28.2 Hz; n = 97 cell-odor pairs; p=0.234; *Figure 2h*), or all activated responses (within experiment mean peak firing rate of all active responses averaged across experiments [lower and upper limit of 95% bootstrapped confidence intervals]: SL, *light-off*: 17.4 Hz [14.6, 20.1], *light-on*: 7.47 Hz [5.49, 9.76]; n = 9 experiments; p=6.51 × 10$^{-5}$, paired t-test; PYR, *light-off*: 14.0 Hz [10.6, 17.7], *light-on*: 12.9 Hz [9.27, 16.6]; p=0.0686; *Figure 2i*).

Although the strength of individual activated PYR responses was unaffected by SL suppression, subtle changes in odor responses, including odor-evoked suppression, may alter the pattern of odor-evoked activity across the population. To quantify changes in the population response pattern of SLs and PYRs, we determined the correlation between trial-averaged population responses for each odor in the *light-on* and *light-off* conditions. Responses were spike count vectors in the first sniff after odor onset for all recorded SLs and PYRs. As a control, we determined the correlation between trial-averaged population responses for each odor in alternating *light-off* trials. SL response patterns across odors were significantly altered in the *light-on* trials compared to the *light-off* (off-off, 0.895 [0.888, 0.901]; on-off, 0.583 [0.526, 0.633]; unpaired t-test, p=5.10 × 10$^{-6}$; *Figure 2j*), while PYR response patterns were unchanged (off-off, 0.834 [0.816, 0.861]; on-off, 0.838 [0.792, 0.876]; unpaired t-test, p=0.882).

We next used a linear support vector machine (SVM) trained and tested on single-trial spike count vectors to classify responses to the panel of presented odors. Odor identity decoding within each condition revealed a substantial decrease in SL performance in the *light-on* condition, with no change in PYR performance (SL, p=0.0116; PYR, p=0.919, paired t-test; *Figure 2k and l*). If anything, in an experiment where SL suppression was complete and decoding performance decreased to chance in the *light-on* condition, decoding in PYRs improved, perhaps due to an increase in the signal-to-noise

ratio in the absence of SL input (**Figure 2k**, bottom). These data indicate that odor responses in PYRs can be accurately classified when SLs are suppressed, but not whether PYR representations remain unchanged across conditions. To address this question, we performed a decoding analysis where the classifier was trained on *light-off* trials and then tested on *light-on* trials. In line with the correlation analysis, we observed a substantial decrease in SL decoding accuracy with no significant change in PYR decoding accuracy (SL, p=0.0035; PYR, p=0.236, paired t-test; **Figure 2m**), indicating that odor-evoked population activity in PYRs is independent of SL input.

## Chemogenetic suppression of SLs does not weaken or reshape PYR odor responses

The mouse PCx is a large structure measuring ~4 mm along its anterior-posterior axis, with long-range intracortical connectivity (**Johnson et al., 2000**; **Franks et al., 2011**; **Hagiwara et al., 2012**). To assuage concerns that our optogenetic SL suppression may be too spatially restricted, we repeated the SL suppression experiment using a chemogenetic approach (**Figure 3a**). AAVs expressing a Cre-dependent inhibitory DREADD, hm4di, were injected at four sites along the anterior-posterior axis of PCx (**Figure 3b**, **Figure 3—figure supplement 1b**). At one of the sites, we co-injected AAVs expressing Cre-dependent Archaerhodopsin (Arch) to opto-tag SLs (fraction of identified SLs, mean ± SD: 0.546 ± 0.146; n = 5 experiments,  four mice, **Figure 3—figure supplement 1a**). Following administration of the DREADD agonists, CNO or Compound 21 (C21), we observed a gradual and marked decrease in SL spontaneous activity and a concomitant small decrease in PYR spontaneous activity (**Figure 3c**).

We examined SL and PYR odor-evoked spike rates before (trials 2–20) and after (trials 41–59) CNO/C21 administration, again only considering cells with activated responses in the baseline condition. As with the optogenetic suppression experiments, odor-evoked response peaks in SLs were lower after CNO/C21 injection, while PYR responses were largely unaffected (**Figure 3d and e**). This was the case for both robustly activated responses (median peak firing rate: SL, baseline: 14.8 Hz, CNO/C21: 6.38 Hz; n = 124 cell-odor pairs; paired t-test, p=0.0086; PYR, baseline: 18.5 Hz, CNO/C21: 17.1 Hz; p=0.0338; n = 96 cell-odor pairs; **Figure 3f**), and for all activated responses (within experiment mean peak firing rate averaged across experiments: SL, baseline: 14.6 Hz [11.6, 19.0], CNO/C21: 10.7 Hz [8.49, 13.3]; n = 5 experiments; paired t-test, p=0.0101; PYR, baseline: 15.1 Hz [10, 22.1], CNO/C21: 12.3 Hz [8.04, 18.1]; p=0.285; **Figure 3g**).

To determine if PYR population odor response patterns were altered after hm4di-mediated suppression of SLs, we calculated the correlation between trial-averaged responses of SLs and PYRs for each odor before and after CNO/C21 injection. As a control, we calculated the correlation between trial-averaged responses for each odor in alternating baseline trials. Responses measured before and after CNO/C21 were markedly different in SLs (baseline-baseline, 0.944 [0.920, 0.953]; CNO/C21-baseline, 0.300 [0.220, 0.393]; p=1.46 × 10⁻⁷, unpaired t-test; **Figure 3h**). PYR responses also changed after CNO/C21 administration (baseline-baseline, 0.950 [0.913, 0.968]; CNO/C21-baseline, 0.729 [0.7, 0.759]; p=2.53 × 10⁻⁶; **Figure 3h**), although this difference was substantially smaller than that observed for SLs (CNO/C21-baseline for SLs versus PYRs, p=9.34 × 10⁻⁶, unpaired t-test). However, unlike the optogenetic suppression experiments in which we compared alternating *light-on* and *light-off* trials, the chemogenetic suppression experiments required comparing responses before and after CNO/C21 had taken effect, which was typically 30 min and 20 trials later.

Therefore, to determine the extent to which the changes we observed after CNO/C21 administration in both SLs and PYRs were directly due to the suppression of SL activity by the activation of hm4di, rather than adaptation, desensitization, attentional changes, or repeated odor exposure (**Jacobson et al., 2018**), we recorded from *Ntng1^Cre* mice that were injected with Cre-dependent mCherry and Arch, for 60 trials without injecting CNO or C21 (fraction of identified SLs, mean ± SD: 0.449 ± 0.146, n = 6 experiments, four mice, **Figure 3—figure supplement 1a and c**). Both SL and PYR responses were decorrelated between the early (trials 2–20) and later trials (trials 41–59), indicating that some changes in response patterns occur with time or following repeated odor presentations (SLs, early-early, 0.952 [0.943, 0.969]; late-early, 0.915 [0.897, 0.934]; p=0.0041; PYRs, early-early, 0.930 [0.910, 0.955]; late-early, 0.849 [0.815, 0.884]; p=6.67 × 10⁻⁴; **Figure 3h**). Additionally, in a separate set of mice injected with AAVs expressing mCherry and Arch, we administered CNO/C21 to account for any nonspecific, hm4di-independent effects of CNO/C21 (fraction of identified SLs, mean ± SD: 0.545 ± 0.229, n

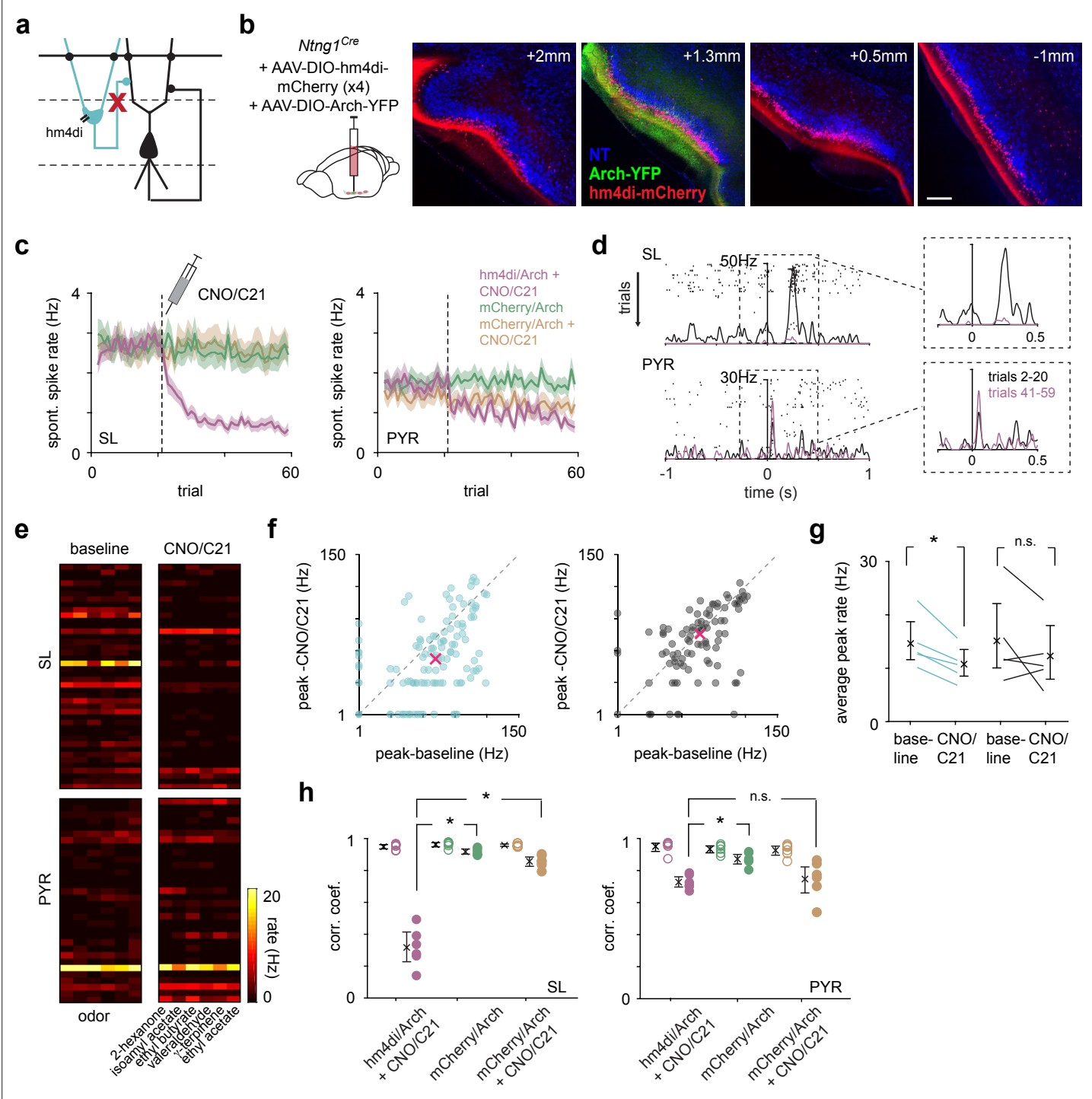

**Figure 3.** Chemogenetic suppression of SLs does not weaken or reshape PYR odor responses. (**a**) SLs were selectively suppressed throughout PCx during recordings using the inhibitory DREADD, hm4di. (**b**) Coronal sections showing selective expression of hm4di-mCherry and Archaerhodopsin (Arch) in SLs. AAVs expressing Cre-dependent hm4di was injected in four sites along the anterior-posterior axis (left to right) of PCx to maximize SL suppression. Arch was co-injected into one of the sites for opto-tagging. Scale bar is 200 µm. (**c**) The average spontaneous spike rate of SLs (left) and PYRs (right) as a function of trials from the experimental and two control groups. CNO/C21 was administered intraperitoneally during the recording, in the 21st trial (dotted line). Confidence intervals are shown. (**d**) Odor response of an example SL and PYR from an experimental mouse (hm4di/Arch + CNO/C21). CNO/C21 trials are shown in magenta and baseline trials in black. Each tick is a spike, and each row of ticks is a trial. (**e**) Heatmaps of trial-averaged spike rates in the first sniff for SL and PYR cell-odor pairs from an example experiment with baseline and CNO/C21 trials plotted separately. (**f**) Odor-evoked response peak firing rates for significantly activated SLs (blue, n = 124 cell-odor pairs) and PYRs (black, n = 96 cell-odor pairs) during

*Figure 3 continued on next page*

*Figure 3 continued*

CNO/C21 trials plotted against baseline trials. Median response peak is shown in magenta. Only the experimental group (hm4di/Arch + CNO/C21) is plotted. (**g**) Mean odor-evoked response peak firing rates within experiments for all activated SLs and PYRs. Each point is an experiment (n = 5). Only the experimental group (hm4di/Arch + CNO/C21) is plotted. (**h**) Correlation coefficients of response vectors between baseline and CNO/C21 trials (filled circles, n = 6 odors) or alternating baseline trials (open circles, n = 6 odors) for each odor for SLs (left) and PYRs (right) for the experimental (magenta) and two control groups (green and orange).

The online version of this article includes the following figure supplement(s) for figure 3:

**Figure supplement 1.** Spread and efficacy of hm4di expression in chemogenetic suppression experiments.

= 3 experiments, three mice). Similar to the previous control, we found that the response patterns changed between the baseline and CNO/C21 conditions in both SLs and PYRs (SLs, baseline-baseline, 0.953 [0.944 0.961]; CNO/C21-baseline, 0.852 [0.824, 0.880]; p=1.23 × 10$^{-4}$; PYRs, baseline-baseline, 0.926 [0.893, 0.949]; CNO/C21-baseline, 0.747 [0.653, 0.823]; p=0.0052; *Figure 3h*). Importantly, the changes in SL responses in the experimental group (hm4di/Arch + CNO) were significantly and substantially greater than the changes in the mCherry/Arch + CNO control group (CNO/C21-baseline for SLs in hm4di/Arch + CNO vs. mCherry/Arch + CNO, p=8.79 × 10$^{-7}$; *Figure 3h*), while changes in PYR responses in the experimental and control groups were comparable (CNO/C21-baseline for PYRs in hm4di/Arch + CNO vs. mCherry/Arch + CNO, p=0.491), indicating that the effect we observed in PYRs was not due to hm4di-mediated suppression of SLs.

In summary, SL responses were substantially degraded following hm4di activation while effects on PYR responses were negligible. Thus, consistent with the optogenetic suppression experiments, PYR odor responses are largely independent of SL input.

## SLs and PYRs have similar response latencies

Next, we determined whether odor-evoked activity in SLs preceded that in PYRs, another prediction of the sequential model. We recorded the odor responses of SLs and PYRs to a panel of 10 odors in a separate set of mice (fraction of optogenetically identified SLs, mean ± SD: 0.486 ± 0.142; n = 15 experiments, 10 mice). We measured the time from inhalation onset to both the response peak (*Figure 4a*) and response onset (i.e., when the odor-evoked spike rate exceeded 2 SD above baseline activity) in a 300 ms time window after inhalation onset for each cell-odor pair (*Figure 4c*). Only activated responses (response index > 0) were analyzed. In both measures, the distributions of response latencies of SLs and PYRs were similar (median latency to peak: SL, 59.0 ms; PYR, 57.0 ms; Wilcoxon rank-sum test: p=0.199; median onset latency: SL, 38.2 ms; PYR, 38.2 ms; Wilcoxon rank-sum test: p=0.587, *Figure 4b and d*). Furthermore, the performance of SLs and PYRs in a classifier trained on spike counts in a 10 or 30 ms sliding window, beginning with inhalation onset, were equivalent (*Figure 4e*). This indicates that odor information accumulates at the same rate in SLs and PYRs. Taken together, our data do not support a sequential activation model in which SL activity precedes PYR activity (limitations of our latency analyses are considered in Discussion).

## Odor response properties of SLs and PYRs

We then asked if and how SLs and PYRs differentially encode odors. If the dense intracortical inputs received by PYRs facilitate the stabilization of odor representations, then PYRs should be more odor selective, their responses less variable across trials, and more discriminable between different odors. We first examined the distributions of activated and suppressed responses of SLs and PYRs, defined using response indices. A value of 1 or –1 indicates that an ideal observer can perfectly discriminate an increase or decrease in firing rate relative to baseline (pre-odor sniff) activity. Both cell types exhibited similar patterns of odor-activated and -suppressed responses but SLs had slightly more suppressed odor responses than PYRs (*Figure 5a and b*), likely due, in part, to their higher baseline firing rates.

We determined the odor selectivity of SLs and PYRs by calculating a lifetime sparseness value for each cell, which describes its firing rate distribution across the panel of odors. A value of 0 means the cell responded identically to all odors, and a value of 1 means the cell responded to only one odor. SLs had significantly lower lifetime sparseness values than PYRs, indicating that SLs are less odor selective than PYRs (mean lifetime sparseness: SL, 0.434 [0.414, 0.458], n = 426 cells; PYR, 0.558 [0.536, 0.581], n = 464 cells; p=2.61 × 10$^{-14}$, unpaired t-test; *Figure 5c*). We also calculated the population sparseness for SLs and PYRs, which describes the firing rate distribution of the population to each odor.

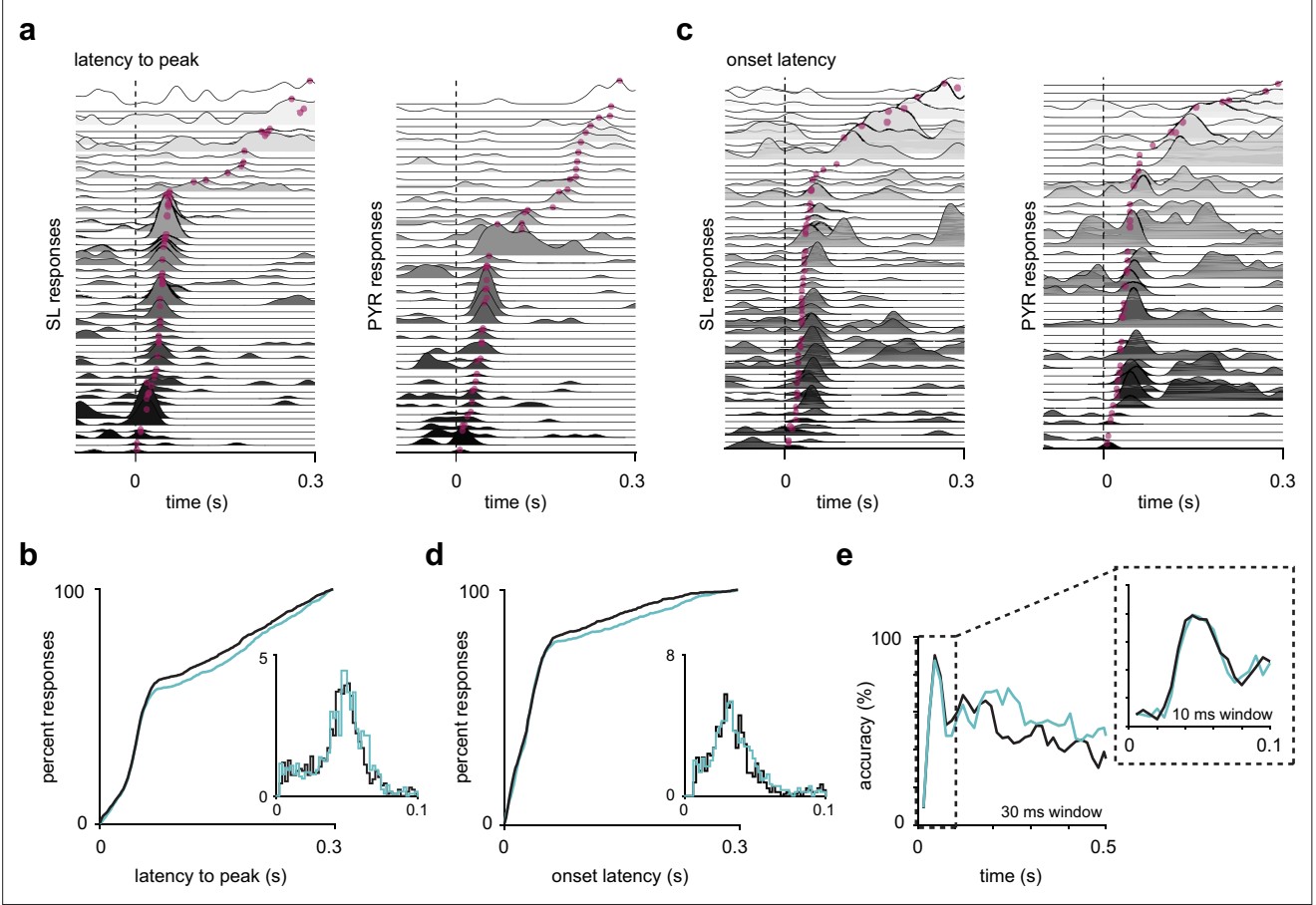

**Figure 4.** Semilunars (SLs) and pyramidals (PYRs) have similar response latencies. (**a**) SL and PYR odor responses from one example experiment sorted by latency to response peak. Dotted line at t = 0 s marks inhalation onset. Magenta dots indicate the time of response peaks. (**b**) Cumulative distribution of latencies to peak for active (response index > 0) SL (blue, n = 1352) and PYR (black, n = 1305) cell-odor pairs. Inset shows histogram of latencies to peak within 100 ms of inhalation onset. (**c**) As in (**a**) but sorted by onset latencies. Magenta dots indicate the time at which each odor response reaches 2 SD above pre-odor baseline. (**d**) As in (**b**) but showing onset latencies for SL (blue, n = 918) and PYR (black, n = 865) cell-odor pairs. (**e**) Odor classification accuracy using a linear support vector machine in a sliding window of either 30 ms or 10 ms (inset).

Here, a value of 0 means all cells contributed equally to the population odor response, and a value of 1 means the population odor response was driven by just one neuron. The SL population response to each odor was significantly less sparse than the PYR population response (SL, 0.686 [0.681, 0.691]; PYR, 0.755 [0.737, 0.771]; n = 10 odors; p=1.61 × 10⁻⁶, unpaired t-test; *Figure 5d*). Together, these data indicate that SLs are less odor selective and their population responses less sparse than PYRs.

We next compared the reliability and discriminability of SL and PYR odor responses. We used z-scored single-trial population response vectors to generate trial-by-trial correlation matrices for SLs and PYRs (*Figure 5e*). We calculated the mean correlation across trials within each odor as a measure of response reliability, and the mean correlation between different odors as a measure of odor discriminability. By this metric, response reliability was similar between SLs and PYRs (SL, 0.474 [0.452, 0.497]; PYR, 0.499 [0.472, 0.525]; n = 10 odors; p=0.191, unpaired t-test), but odors were slightly more discriminable in PYR responses (SL, 0.266 [0.254, 0.279]; PYR, 0.238 [0.220, 0.253]; p=0.0193; *Figure 5f*).

To further investigate the discriminability of odor responses, we performed principal components analysis (PCA) and, for visualization, projected SL and PYR z-scored pseudopopulation responses onto their first three principal components for each odor (*Figure 5g*). We quantified the Euclidean distances between the means of the first six principal components of pairs of odors and found that odors were more separable in PYR responses than in SL responses (median distance: SL, 19.4; PYR, 27.4; n = 45 odor pairs; p=3.21 × 10⁻⁷, unpaired t-test; *Figure 5h*). In PYRs but not SLs, odor responses

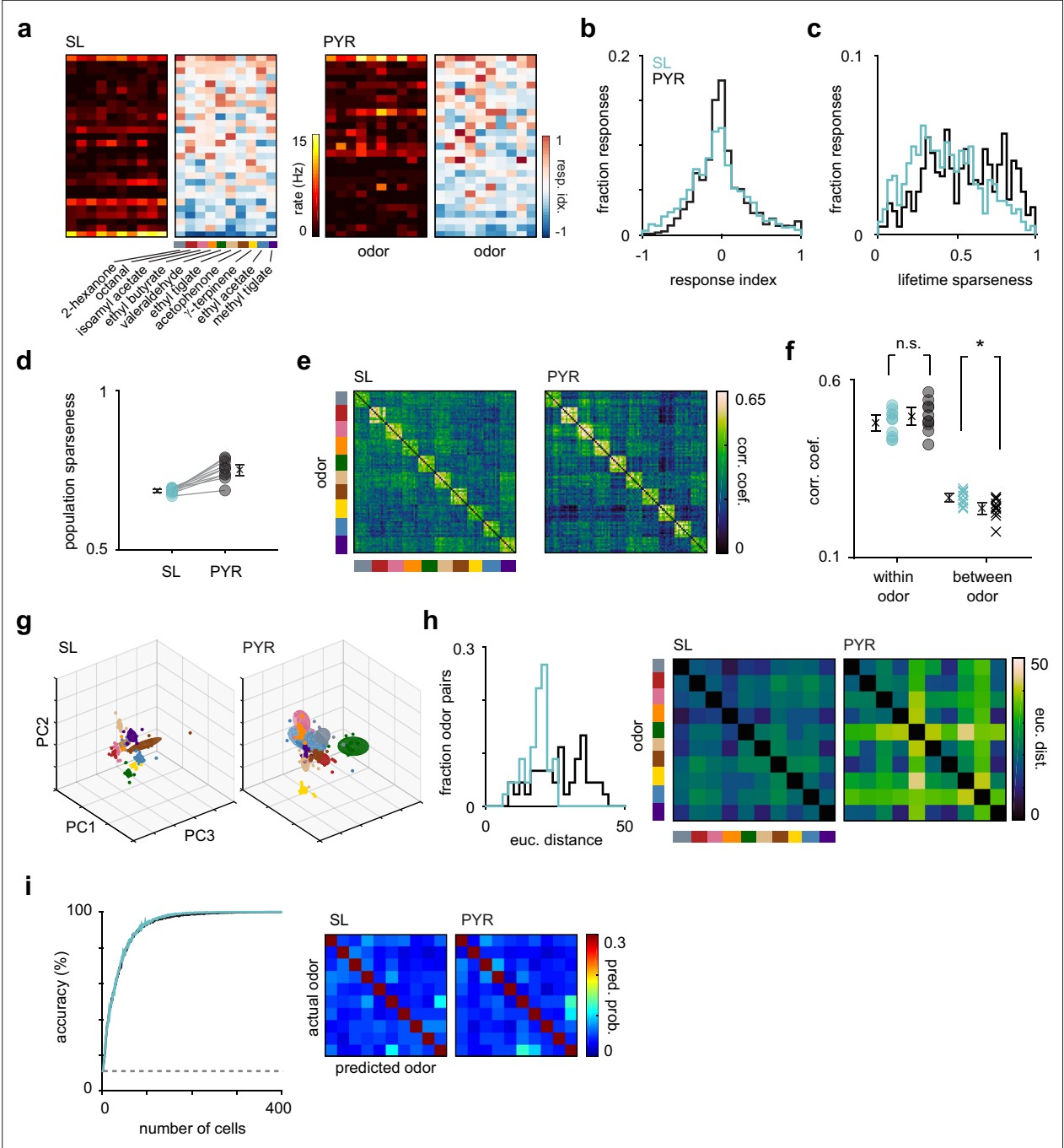

**Figure 5.** Odor response properties of semilunars (SLs) and pyramidals (PYRs). (**a**) Heatmaps of spike rates in the first sniff and response indices for cell-odor pairs from an example experiment for SLs and PYRs sorted by number of activated responses. (**b**) Distribution of response indices for SL (blue, n = 4260) and PYR (black, n = 4640) cell-odor pairs across all experiments. (**c**) Distribution of lifetime sparseness for SLs (blue, n = 426 cells) and PYRs (black, n = 464 cells). (**d**) Population sparseness for SL and PYRs. Each circle is the average across experiments for an odor (n = 10 odors). (**e**) Trial-by-trial z-scored spike count correlation (Pearson's) matrices sorted by odor for SLs and PYRs. Correlation matrices were generated for each experiment individually and then averaged across experiments. (**f**) Mean correlation across trials within each odor (circles) for SLs (blue) and PYRs (black) and between odors (crosses). (**g**) Principal components analysis was performed on z-scored SL (left) and PYR (right) pseudopopulation responses. Responses were projected onto the first three principal components. Each colored sphere represents an odor centered on the mean and encompasses one standard deviation. Colored dots indicate individual trials. (**h**) Distribution of Euclidean distances between the mean of the first six principal components for pairs of odors for SLs (blue) and PYRs (black). Distances are shown as confusion matrices on the right. (**i**) SL (blue) and PYR (black) odor classification accuracy determined using a linear support vector machine as a function of pseudopopulation size. The decoder was asked to classify responses on a single trial to the panel of 10 odors. Classification accuracy was averaged across 100 iterations for each pseudopopulation size. Dashed line indicates

*Figure 5 continued on next page*

*Figure 5 continued*

chance performance. Confusion matrices, averaged across 100 iterations, on the right show the probability that an odor was predicted.

The online version of this article includes the following figure supplement(s) for figure 5:

**Figure supplement 1.** Optogenetic suppression of semilunars (SLs) does not change pyramidal (PYR) odor response properties.

to valeraldehyde, γ-terpinene, and ethyl acetate were more distinct than their responses to the other presented odors (*Figure 5e and g*), resulting in a larger spread in distances between pairs of odors in PYRs. Finally, we asked if a downstream decoder could more accurately identify odors based on PYR versus SL odor-evoked activity. However, odor classification accuracy, at all pseudopopulation sizes, was similar in SLs and PYRs (*Figure 5i*, left).

To summarize, odors were slightly more discriminable in PYRs than in SLs, consistent with the idea that the recurrent network decorrelates PYR responses to different odors. However, the trial-to-trial variability and the odor decoding performance of SLs and PYRs were equivalent. We additionally analyzed if and how the odor response properties of PYRs changed when SLs were suppressed and found no substantial effects, further suggesting that SLs and PYRs encode odor information in parallel (*Figure 5—figure supplement 1*).

If both SL and PYR responses derive solely from their sensory inputs, then similar odors should evoke similar responses in both cell types. Interestingly, however, we noticed some subtle but distinct differences in the off-diagonal structure of the correlation matrices for SL and PYR responses as well as in the confusion matrices of pairwise Euclidean distances between odors (*Figure 5e and h*). For example, valeraldehyde and γ-terpinene responses were more strongly decorrelated from other odor responses in PYRs than in SLs. Also, methyl tiglate responses were more like ethyl tiglate than acetophenone in SLs, while the reverse was true in PYRs. These differences were similarly reflected in the confusion matrices showing predicted odors when decoding using just 25 randomly selected cells for SLs and PYRs (scale adjusted to emphasize classification errors) (*Figure 5i*, right). These observations indicate that SLs and PYRs are transforming odor information from the OB differently, and that their outputs do not simply reflect their sensory inputs.

## Discussion

The *Ntng1^{Cre}* mouse line allowed us to distinguish between SLs and PYRs and selectively suppress SLs in extracellular recordings of layer II PCx neurons in awake, head-fixed mice. We found that odor responses in SLs and PYRs are largely independent as suppressing SLs, both optogenetically and chemogenetically, did not weaken or alter the response pattern of PYRs. We additionally did not find differences in odor response latencies between SLs and PYRs. These results are not consistent with a sequential processing model in which SLs are the primary cortical recipients of OB input whose activity is required to drive PYR responses. We found that SLs are more broadly tuned, less sparse, and their population odor responses less discriminable compared to PYRs. Taken together, our data support a parallel processing model in which OB input drives both SL and PYR odor responses, which then perform different computations on this information before routing it to different downstream areas.

### Stimulus selectivity, discriminability, and response variability

Individual PCx neurons integrate convergent inputs from different combinations of glomeruli (*Apicella et al., 2010*; *Davison and Ehlers, 2011*). SLs almost exclusively receive afferent OB inputs, while PYRs receive both afferent OB inputs and dense intracortical inputs from SLs and other PYRs (*Johnson et al., 2000*; *Franks et al., 2011*; *Suzuki and Bekkers, 2011*; *Hagiwara et al., 2012*). If intracortical connectivity is random, then sensory information from the OB received by PCx neurons would be uniformly redistributed among PYRs via the recurrent network. Individual PYRs would therefore effectively sample a far larger subset of glomeruli than individual SLs, resulting in more broadly tuned and overlapping responses among PYRs (*Suzuki and Bekkers, 2011*). Alternatively, coordinated activity within PCx could actively sculpt the recurrent circuitry so that PYRs that are regularly co-activated, because they respond to similar chemical features, become preferentially interconnected, resulting in more narrowly tuned, reliable, and discriminable responses than those in SLs (*Haberly, 2001*). Our results are largely consistent with the latter model in that we found PYR responses to be more narrowly

tuned and discriminable than those in SLs. A recent study showed that odor responses in layer III of PCx, which only contains PYRs, were more correlated between chemically similar odor stimuli than responses in layer II, which contain both SLs and PYRs, or the OB (*Pashkovski et al., 2020*), providing further evidence for structured associative inputs. It was therefore surprising that PYR responses were not more reliable than SL responses.

Our findings on the stimulus selectivity and discriminability of SLs and PYRs are also comparable to those reported in two zebrafish odor centers downstream of the OB: cells in the ventral telencephalon (Vv) receive primarily OB inputs and exhibit broader stimulus tuning and more overlapping odor representations than those in the dorsal telencephalon (Dp), which is the teleost homolog of PCx (*Yaksi et al., 2009*). Finally, in the control experiments for the DREADD suppression of SLs, we noticed that PYR responses seemed to change more with time and/or stimulus presentation than SL responses. This result is consistent with an activity-dependent sculpting of recurrent connectivity so that PYR responses evolve while SL responses continue to faithfully represent the stimulus (*Jacobson et al., 2018*; *Bolding et al., 2020*; *Pashkovski et al., 2020*; *Schoonover et al., 2021*).

## Interpreting effect size

Although the response properties of SLs and PYRs are consistent with a more sensory and a more associative network, respectively, the differences we observed were less pronounced than we expected. There are several possible explanations for this. First, although SLs and PYRs are the two main classes of principal neurons in layer II, some neurons have intermediate morphological and intrinsic properties that lie on a gradient from canonical SLs to canonical PYRs (*Yang et al., 2004*; *Suzuki and Bekkers, 2011*; *Wiegand et al., 2011*). These cells, which may or may not be Ntng1+, could be diluting the differences that we observed. Additionally, although we have used multiple stringent selection criteria (*Figure 1—figure supplement 5*) to reliably identify SLs, and have omitted experiments with a low fraction of opto-tagged cells, it may still be possible that some SLs have been miscategorized as PYRs and vice versa. Second, although SLs receive stronger OB inputs than PYRs, this difference is relative and does not mean that OB inputs onto PYRs are weak or that PYRs are not directly driven by the OB, as this is sometimes interpreted. Third, unlike in slice recordings, the recurrent network and top-down inputs are preserved in in vivo recordings. Therefore, while SL activity is driven largely by sensory input, PYR activity is also influenced by ongoing background activity that provide non-olfactory inputs to PYRs, rendering their responses to odors more variable and less discriminable than might be expected (*Stringer et al., 2019*). Why did we find no difference in the response latencies of SLs and PYRs even though SLs responded approximately 2–4 ms earlier than PYRs in response to LOT stimulation in brain slice experiments or in anesthetized rats (*Ketchum and Haberly, 1993*; *Suzuki and Bekkers, 2011*; *Wiegand et al., 2011*)? In vivo, ongoing recurrent and top-down inputs in awake animals may depolarize PYRs and thereby shorten their response latencies. More importantly, electrical LOT stimulation generates a single, synchronous barrage of afferent inputs into PCx. However, different mitral cells respond to odors with variable latencies that tile the entire ~300 ms sniff cycle (*Cury and Uchida, 2010*; *Shusterman et al., 2011*; *Bolding and Franks, 2018*), making it difficult to compare SL and PYR onset latencies to a given odor-activated input. Precise optogenetic stimulation of OB glomeruli with high temporal precision (*Chong et al., 2020*) can be used to address this question more directly.

## Potential limitations of optogenetic and chemogenetic suppression

With optogenetics, we were able to effectively suppress the SLs we were recording from on alternating trials, thus providing a robust and ongoing comparison of PYR responses with and without SL input. However, PCx spans ~4 mm along its anterior-posterior axis and is a long-range recurrent network with neurons forming intracortical connections with neurons >1 mm away (*Johnson et al., 2000*; *Franks et al., 2011*; *Hagiwara et al., 2012*). Hence, to effectively remove SL inputs onto PYRs, we needed to suppress SLs located far from the recording site. However, the spatial resolution of optogenetic suppression using 532 nm light, determined using a photobleaching assay, is slightly less than 1 mm (*Li et al., 2019*). To broaden the area of suppression, we used a thicker (400 μm) optic fiber and Jaws, a red-light (635 nm)-activated opsin, as red light scatters less in tissue (*Chuong et al., 2014*; *Wiegert et al., 2017*). We also injected the virus in 3–4 sites along the anterior-posterior axis of PCx to suppress axon terminals, at the recording site, of remote cell bodies outside

of the light cone. Nevertheless, it is possible that we failed to disrupt PYR responses in *light-on* trials simply because we were not suppressing enough SLs across PCx. If so, we might expect that more complete SL suppression would decrease PYR responses and impair their odor decoding. However, odor decoding accuracy in PYRs trended upward on *light-on* trials in experiments with robust local SL suppression, indicating that SL suppression did affect PYRs, and suggesting that SLs were suppressed well beyond the recording area. Chemogenetics alleviates the spatial limitations of optogenetics, but the suppression of odor-evoked activity in SLs was less robust than in the optogenetic experiments. Additionally, we had to wait 20–30 min for cells to be suppressed, which allowed for shifts in odor representations. Novel techniques that enable more effective axon terminal suppression with high spatiotemporal precision and efficacy may enable a more robust interrogation of the effects of suppressing SLs on PYRs. These caveats notwithstanding, both the optogenetic and chemogenetic experiments showed effective SL suppression and indicated that SL suppression had little to no effect on PYR odor responses.

## Roles of SLs and PYRs in odor processing

Our data do not support a two-stage sequential activation model, where odor information from the OB is transmitted first to SLs and then from SLs to PYRs. Instead, both SLs and PYRs can be driven by OB input and appear to differentially process odor information, as indicated by their distinct tuning properties and by the differences we observed in the off-diagonal structure of their response correlation matrices. Interestingly, SLs preferentially project to the posterior PCx, posteromedial cortical amygdala, and lateral entorhinal cortex while PYRs project back to the OB, medial prefrontal and orbitofrontal cortices (*Diodato et al., 2016*; *Mazo et al., 2017*). Because SLs are less driven by intracortical inputs, downstream regions receiving SL inputs may receive odor information that more closely reflects the stimulus. By contrast, PYRs integrate afferent OB input with inputs from SLs and other PYRs, as well as inputs from other brain areas to form contextual odor representations and provide a substrate for the storage of odor representations. Thus, the varying ratios of afferent versus associative connectivity between SLs and PYRs perhaps enable each cell type to extract distinct features of odor information and route this information to different downstream regions.

Although it may not be surprising that PYRs are able to respond to odors without SL input, our finding that PYR odor representations are essentially equivalent with and without SL input is nevertheless puzzling. Why might this be and what does this imply about the function of SL projections onto PYRs? Inputs onto PYRs from other PYRs are approximately five times stronger than inputs from SLs (*Figure 1—figure supplement 3e and h*). It might be that under our very simplified conditions, with pure odors delivered at relatively high concentrations (0.3% vol./vol.), the contribution from SLs becomes insignificant and may be more apparent in more naturalistic odor environments. Alternatively, in the optogenetic experiments, we suppressed SLs hundreds of milliseconds before odor onset, which could be sufficient time for the network to re-equilibrate (*Bolding and Franks, 2018*). Perhaps rapid suppression of SLs during or immediately before the odor would have a more pronounced effect. We also presented odors passively, in nonbehaving mice, and it is possible that differences between, and dependencies on, the different cell types might emerge in mice engaged in a behavioral task (*Chen et al., 2015*). However, this seems unlikely given recent evidence that odor representations in PCx do not appear to encode odor value or be task-specific (*Gadziola et al., 2020*; *Millman and Murthy, 2020*; *Wang et al., 2020*) and that any 'top-down' effects would almost certainly influence PYR output more than SL. Finally, perhaps SLs and PYRs do not simply represent odor identity and we are looking at the wrong feature of the odor response. Future experiments will be required to resolve this, but our data make clear that odor processing in PCx does not occur through a two-stage process mediated first by SLs and then by PYRs.

## Parallel loops in PCx and hippocampus

Although we have proposed a parallel model for processing odor information in PCx, it is not strictly correct as PYRs receive input from both OB and SLs. This organization may be better described as a parallel-loop motif, as in the hippocampus. PCx and the hippocampus are analogous structures. Both are trilaminar paleocortices with extensive recurrent networks (*Guzman et al., 2016*) that support unsupervised learning (*Barkai et al., 1994*; *Haberly, 2001*; *Rolls, 2013*). In terms of morphology and connectivity, PYRs are analogous to CA3 pyramidal cells and SLs to dentate gyrus (DG) granule cells.

The roles of DG and CA3 in the encoding of episodic memories are relatively well characterized; DG granule cells are thought to drive the acquisition of memories, but not retrieval, and perform pattern separation while CA3 pyramidal cells are thought to store and retrieve memories and perform pattern completion (*Leutgeb et al., 2007*; *Neunuebel and Knierim, 2014*; *Madar et al., 2019*; *Hainmueller and Bartos, 2020*). It is important to note that although DG granule cells synapse onto CA3 pyramidal cells, the DG-CA3 circuit is not exclusively sequential. CA3 also receives direct input from the entorhinal cortex, and when DG projections to CA3 are silenced or ablated, the ability of CA3 to retrieve memories is unimpaired (*Lee and Kesner, 2004*). In our previous study with the *Ntng1^Cre* mouse line, we found that PYRs were more robust to degraded afferent OB inputs and exhibited more persistent odor-evoked activity after stimulus offset than SLs (*Bolding et al., 2020*). Both features are consistent with an associative network performing computations that stabilize and enable reliable retrieval of odor ensembles, characteristic of a pattern completing network (*Rolls, 2013*; *Inagaki et al., 2019*). However, there is yet to be evidence implicating SLs, specifically, as pattern separators in the olfactory circuit (*Chapuis and Wilson, 2011*). A key property of DG granule cells that enable them to effectively decorrelate similar activity patterns is their sparseness (*Leutgeb et al., 2007*). However, we found that SLs had higher spontaneous spike rates, were individually more broadly tuned, and their population response less sparse than PYRs.

## Conclusion

In conclusion, our data do not support a sequential processing model for PCx in which odor information from OB is first integrated in SLs, and then routed from SLs to PYRs. Instead, we propose that these distinct excitatory cell types form largely parallel processing schemes, where each cell type receives varying amounts of sensory and associative inputs, performs distinct local computations, and projects the information to distinct downstream regions.

# Materials and methods

All experimental protocols were approved by the Duke University Institutional Animal Care and Use Committee. Information about the *Ntng1^Cre* mouse line is described in *Bolding et al., 2020*. Methods for in vivo extracellular recording and preliminary analysis have been reported in detail in *Bolding and Franks, 2017* and *Bolding et al., 2020* and are summarized here.

## Subjects

All experiments were performed on adult (6–10 weeks old) *Ntng1^Cre* knock-in mice, which were generated by inserting the DNA sequence encoding Cre-recombinase at the start codon of the *Ntng1* gene using CRISPR. All mice were heterozygotes, crossed to either Ai14 (B6.Cg-Gt(ROSA)26Sor^tm14(CAG-tdTomato)Hze/J, 007914) or C57BL/6J (000664) mice obtained from The Jackson Laboratory.

## Statistics

Unless otherwise stated, data are reported as mean [LL UL], where LL and UL are the lower and upper bounds of 95% confidence intervals for the mean, respectively. Confidence intervals were calculated by bootstrapping the data 1000 times and calculating the mean for each sample using MATLAB's 'bootci' function. Statistical significance of comparisons between groups was determined using MATLAB's 'ttest,' 'ttest2,' or 'ranksum' functions.

## Stereotaxic injections and headpost placement

Mice were given buprenorphine-SR (0.1 mg/kg, s.c.) 10 min before the start of surgery and then placed in a closed glass box filled with 5% isoflurane to induce anesthesia. Afterward, they were moved to a stereotaxic frame and maintained on isoflurane anesthesia (0.5–2% in 0.8 L/min O$_2$) throughout the procedure. A local anesthetic, bupivacaine, was injected under the skin over the skull and an incision was made along the midline to expose the skull. Between 1 and 4 burr holes, depending on the experiment, were made over PCx using coordinates relative to bregma. AAVs (either AAV5-hSyn-DIO-GFP, AAV8-hSyn-DIO-Jaws-eYFP or AAV5-hSyn-DIO-ArchT3.0-eYFP or AAV5-hSyn-DIO-hm4di-mCherry or AAV5-hSyn-FLEX-mCherry, all obtained from UNC Vector Core) were then delivered through a glass pipette using a Nanoject pump (Drummond) (200 nL/site at 60 nL/min). The following coordinates

were used (in mm, AP 2.0, ML 2.15, DV 3.8; AP 1.3, ML 2.82, DV 3.95; AP 0.5, ML 3.5, DV 4.05; AP –0.6, ML 3.8, DV 4.10). After slowly retracting the pipette, burr holes were covered using KwikCast sealant and a custom titanium headpost was lowered onto the exposed skull and attached using Metabond (Parkell, Inc). Mice were placed on a heating pad for 30 min after surgery while they recovered from anesthesia.

## In vitro electrophysiology and analysis

Mice were anesthetized with isoflurane and decapitated, and the cortex was quickly removed in ice-cold artificial CSF (aCSF). Parasagittal brain slices (300 μm) were cut using a vibrating microtome (Leica) in a solution containing (in mM): 10 NaCl, 2.5 KCl, 0.5 CaCl2, 7 MgSO4, 1.25 $NaH_2PO_4$, 25 $NaHCO_3$, 10 glucose, and 195 sucrose, equilibrated with 95% $O_2$ and 5% $CO_2$. Slices were incubated at 34°C for 30 min in aCSF containing 125 mM NaCl, 2.5 mM KCl, 1.25 mM $NaH_2PO_4$, 25 mM $NaHCO_3$, 25 mM glucose, 2 mM $CaCl_2$, 1 mM $MgCl_2$, 2 NaPyruvate. Slices were then maintained at room temperature until they were transferred to a recording chamber on an upright microscope (Olympus) equipped with a ×40 objective.

For current-clamp recordings, patch electrodes (3–6 megohm) contained 130 K methylsulfonate, 5 mM NaCl, 10 HEPES, 12 phosphocreatine, 3 MgATP, 0.2 NaGTP, 0.1 EGTA, 0.05 AlexaFluor 594 cadaverine. For voltage-clamp experiments, electrodes contained 130 D-gluconic acid, 130 CsOH, 5 mM NaCl, 10 HEPES, 12 phosphocreatine, 3 MgATP, 0.2 NaGTP, 10 EGTA, 0.05 AlexaFluor 594 cadaverine. Voltage- and current-clamp responses were recorded with a MultiClamp 700B amplifier, filtered at 2–4 kHz, and digitized at 10 kHz (Digidata 1440). Series resistance was typically ~10 megohm, always <20 megohm, and was compensated at 80–95%. The bridge was balanced using the automated MultiClamp function in current-clamp recordings. Data were collected and analyzed offline using AxographX and IGOR Pro (Wavemetrics). Junction potentials were not corrected.

## Intrinsic properties

Recordings were targeted to SLs (tdTomato+) and PYRs (unlabeled) located in layer II. Cells were also visualized using a fluorescent indicator (Alexa 594 Cadaverine) to confirm that they were correctly identified as SL or PYR. Intrinsic properties were measured at current clamp in response to a series of 1-s-long current pulses stepped up in 50 pA increments.

## Synaptic connectivity

All experiments were performed 3–4 weeks after virus injection (AAV-hSyn-DIO-Chr2 or AAV-hSyn-ChR2). We recorded from uninfected SLs (tdTomato+) and PYRs (unlabeled) adjacent to the infection site. To ensure cells were uninfected, we first examined responses to weak, 1-s-long light pulses (470 nm, CoolLED) delivered through the ×40 objective. Cells that exhibited large and sustained photocurrents were discarded. Uninfected cells were held at either –70 mV or +5 mV to isolate excitatory or inhibitory synaptic currents, respectively. Brief (1 ms, ~10 mW) pulses were delivered every 10 s to activate ChR2+ axon terminals.

## In vivo extracellular recordings

### Electrode placement and data acquisition

Recordings were performed 3–6 weeks after virus injections. On the day of the recording, the Metabond and KwikCast covering the craniotomy made during virus injection was removed. A 32-site polytrode acute probe, either with a 50 μm optic fiber attached (A1x32-Poly3-5mm-25s-177-OA32LP, Neuronexus, Ann Arbor, MI) or without (A1x32-Poly3-5mm-25s-177-A32, Neuronexus) was positioned over the craniotomy and lowered into one of the virus injection sites using a Patchstar Micromanipulator (Scientifica, UK). No additional craniotomies were made on the day of the recording, thus mice were never anesthetized. Recordings were targeted to 3.8–4.3 mm ventral to the brain surface, and the probe was lowered until a band of intense spiking activity, reflecting the densely packed layer II of PCx, was observed. Electrophysiological signals were acquired through an A32-OM32 adaptor (Neuronexus), digitized at 30 kHz at a CerePlex digital headstage (Blackrock Microsystems, Salt Lake City, UT) and recorded using a Cerebus multichannel data acquisition system (Blackrock Microsystems). Experimental events (e.g., odor delivery times, laser pulse times) and respiration signals were

acquired at 2 kHz by analog inputs of the Cerebus system. Respiration was monitored using a micro-bridge mass airflow sensor (Honeywell AWM3300V) positioned opposite the animal's nose.

## Spike sorting

Individual units were isolated using Spyking-Circus (https://github.com/spyking-circus) (*Yger et al., 2018*). Isolated clusters were manually curated to remove those with >1% of interspike intervals (ISIs) violating the refractory period (<2 ms) or that looked like noise artifacts. Pairs of clusters with similar waveforms and coordinated refractory periods in the cross-correlograms were merged into a single cluster. The odor response recording session and the opto-tagging recording session for each experiment were merged and sorted as one session and later split for subsequent analysis.

## Opto-tagging

Opto-tagging was performed at the end of each experiment (or at the start for experiments using DREADDs). When the probe without the attached optic fiber was used, a separate 200 μm or 400 μm optic fiber (ThorLabs) was implanted at a 12° angle to the probe, and a distance of 500–600 μm away from the probe's ventral coordinate. The optic fiber was connected to a mechanical shutter (Uniblitz, Vincent Associates, Rochester, NY) and then to either a 532 nm or 635 nm laser (OptoEngine LLC, Midvale, UT) using a patch cable (ThorLabs). The mechanical shutter was controlled using TTL pulses sent from an analog output of the Cerebus system, which then allowed light pulses through the optic fiber. A series of 1000 pulses (300 ms, 1 Hz) were delivered to opto-tag cells. A limitation of using inhibitory opto-tagging is that spontaneously low-firing Ntng1+ cells may not be identified. To alleviate this issue, we delivered a high number of light pulses and recorded for ~18 min to ensure the spiking activity of spontaneously low-firing cells was sufficiently sampled. The same TTL pulse was also sent to an analog input channel on the Cerebus system to align light pulses with electrophysiological recordings post hoc. Another limitation of opto-tagging is we may incorrectly identify Jaws- cells that are synaptically connected to Jaws+ cells as light-responsive. We therefore added a selection criterion based on the time course of suppression of each cell. Latency to suppression was determined by change point analysis (*Wolff et al., 2014*; *Rowland et al., 2018*). PSTHs were generated for each unit using 2 ms bins spanning 100 ms before to 100 ms after laser onset, and a change point within that time was determined using MATLAB's 'findchangepts' function. A cell was categorized as light-responsive if (1) its trial-averaged baseline activity was significantly higher than that during the laser pulse (Wilcoxon rank-sum test, $p < 1 \times 10^{-7}$), (2) the area under the receiver operating characteristic (auROC) curve was less than 0.5, and (3) the change point value was lesser than 10 ms from the laser on time. In 4/9 optogenetic suppression experiments, we delivered 60 light pulses that were triggered by a TTL pulse to the laser (i.e., no shutter), and thus we were not able to accurately determine change points. For these experiments, light-responsive cells were determined using the rank-sum test and auROC criteria, followed by manual curation. Units that exhibited fast suppression were categorized as light-responsive, while units that exhibited visibly slow or delayed suppression were categorized as non-light responsive.

## Odor stimuli

The odor stimulus set consisted of 2-hexanone (Sigma, 02473), octanal (Aldrich, O5608), isoamyl acetate (Tokyo Chemical Industries, A0033), ethyl butyrate (Aldrich, E15701), valeraldehyde (Aldrich, 110132), ethyl tiglate (Alfa Aesar, A12029), acetophenone (Fluka, 00790), γ-terpinene (Aldrich, 223190), ethyl acetate (Sigma-Aldrich, 34858), and methyl tiglate (Alfa Aesar, A11964).

## Odor delivery

Odors were delivered using a custom-built 16-valve olfactometer. House air was passed through a filter and then split three ways into three mass flow controllers (MFCs) (Aalborg, Orangeburg, NY). MFC1 flowed filtered air at a constant rate of 1 L/min while airflow through MFC2 and MFC3 were varied to vary the concentration of odors delivered while maintaining a total airflow out of the olfactometer at 1 L/min. Custom Arduino software was used to control odor solenoid opening and closing as well as to log which odor was delivered at each trial. Monomolecular odorants were diluted to 1% v./v. in mineral oil. Normally, a 1 L/min clean air stream from MFC1 was directed to the mouse's nose. During a trial, air from either MFC2 or MFC3 was directed through one of the odor vials and the

odorized air stream was directed to exhaust for an equilibration period of 6 s before rapid switching of a final valve, triggered on exhalation, redirected the odorized air to the nose and neutral air to exhaust. Odors or mineral oil blank control stimuli were presented for 1 s, and stimuli were presented every 10 s in random order. Odors were presented at 0.3% air dilution for the optogenetic and chemogenetic suppression experiments and at 1% for analyzing odor response latencies and properties.

## Spontaneous firing rate

Spontaneous firing rates in *Figure 1l* were determined by counting the number of spikes in a 4 s inter-trial window starting 4 s after odor offset. Spontaneous firing rates in *Figures 2e and 3c* were determined by counting the number of spikes in a 500 ms window before odor presentation.

## Individual cell-odor responses

We omitted the first-odor trial in all analyses (*Jacobson et al., 2018*). Individual trial-averaged cell-odor responses as a function of time were visualized by computing kernel density functions (KDFs) smoothed using a 10 ms Gaussian kernel (Chronux Toolkit). The reliability of responses was determined using the auROC metric. auROC values were computed from the U-statistic of the rank-sum test comparing spike counts of each cell-odor pair during the first sniff after odor presentation to spike counts during the sniff preceding odor presentation. auROC values were then converted to response indices using the formula: auROC * 2 – 1. A value of –1 or 1 indicates a suppressed or activated odor response that was perfectly distinguishable from the pre-odor baseline response while a value of 0 indicates no difference between the odor and pre-odor baseline response.

To determine response peaks, KDFs were first computed for each cell-odor pair and the maximum value within the specified time window was then determined as the peak. Significantly active cell-odor pairs were ones with a response index > 0 and a p-value<0.05 (Wilcoxon rank-sum test) comparing spike counts in the first sniff with spike counts in the pre-odor sniff. Responses where no peaks were found within the specified time window were given a value of 0 for *Figures 2h and 3f*.

## Latency analysis

Response latencies were determined for all activated cell-odor pairs (response index > 0). For peak latencies, KDFs were first computed from spike counts occurring in a 300 ms time window from inhalation onset using a 10 ms Gaussian kernel. Then, the time of the maximum value of the KDF was identified and taken as the latency to peak. For onset latencies, KDFs were similarly computed but for a time window of 600 ms, spanning 300 ms before inhalation onset (baseline) and 300 ms after. The mean firing rate and standard deviation were determined for the baseline period, and the time after inhalation onset at which the KDF first exceeded 2 SDs above the baseline was then taken as the onset latency.

Odor classification accuracy in a sliding window was determined using a linear multiclass SVM classifier using leave-one-out cross-validation, as described in the 'Population decoding analysis' section. Within each time window, the pseudopopulation spike count vector for one trial was omitted, the classifier trained on the remaining trials, and the odor presented during the omitted trial was predicted using the trained SVM. Classification within each window was repeated 100 times, and accuracies for each window are the average across all iterations.

## Sparseness

Lifetime sparseness describes the firing rate distribution of individual neurons to all of the odors presented and reflects the tuning width of neurons. A value of 1 indicates the cell responded to just one odor, and a value of 0 indicates the cell's firing rate was distributed uniformly across all odors. Population sparseness is a measure of the density of responses of a population of cells, where a value of 1 indicates the population response to a given odor was driven by just one cell and a value of 0 indicates uniform contribution of all neurons in the population to a given odor response. Lifetime and population sparseness values were calculated for each experiment individually:

$$\text{lifetime sparseness} = \frac{1 - (\sum \frac{r_i}{n})^2 / \sum \frac{r_i^2}{n}}{1 - \frac{1}{n}}$$

where $r_i$ is the trial-averaged response to the $i$th odor and n is the number of odors. Population sparseness for each odor was calculated with the same formula but $r_i$ is now the trial-averaged response of the $i$th cell and n is the number of cells.

## Trial-trial population vector distances and principal components analysis

The similarity of population odor responses, which are vectors of z-scored spike counts in the first 500 ms after inhalation onset, was determined by correlating response vectors from pairs of trials in each experiment separately. Spike counts were z-scored to a 500 ms epoch before inhalation onset (–0.6 to –0.1 s before inhalation onset [t = 0]). The mean within-odor correlation was calculated by averaging correlation coefficients of trials from the same odor while the mean between-odor correlation was calculated by averaging correlation coefficients of all trials except those of the same odor. PCA was performed using MATLAB's built-in 'pca' function (singular value decomposition algorithm) by using z-scored pseudopopulation (responses from all experiments were pooled together) spike count vectors.

## Population decoding analysis

Odor classification accuracy was determined with a linear multiclass SVM classifier using leave-one-out cross-validation (LIBLINEAR, solver 4 –support vector classification by the Crammer and Singer method, https://www.csie.ntu.edu.tw/~cjlin/liblinear/). The feature vectors used in the classifier in *Figure 5* were pseudopopulation vectors of raw spike counts in the first 500 ms after inhalation onset. To determine classification accuracy at increasing numbers of cells, populations of varying sizes were constructed by subsampling the total pseudopopulation 100 times and averaging the classification accuracy across all 100 iterations for that population size. Decoding in individual experiments as in *Figures 2 and 3* was performed using the same decoder but with size-matched populations of SLs and PYRs from each experiment.

## DREADD agonists dosage and delivery

250 µL of either CNO (4 mg/kg, Sigma, C0832) or compound 21 (1 mg/kg, Sigma, SML2392) diluted in sterile 0.9% saline solution was injected intraperitoneally during the recording session.

## Immunohistochemistry

Mice were anesthetized with isoflurane and then perfused transcardially with cold 4% paraformaldehyde in 0.2 M phosphate buffer (4% PFA). Brains were dissected and post-fixed for 48 hrs in 4% PFA at 4°C and then placed in a vibrating microtome, submerged in cold 1× phosphate buffered saline (PBS, Sigma, P4417) to slice 50 µm sections of OB and PCx. To stain for GFP and mCherry, sections were permeabilized using 0.1% Triton X-100 (Acros Organics, 327371000) in 1× PBS (t-PBS) for three washes and then with 0.3% t-PBS for one wash. Slices were incubated with chicken anti-gfp (1:400, Invitrogen, A10262), rabbit anti-rfp (1:400, Rockland, 600-401-379) in blocking buffer (5% normal goat serum; EMD Millipore, S26) in 0.3% t-PBS overnight at 4°C. The next day, slices were rinsed in 0.1% t-PBS for three washes and then incubated in AlexaFluor 488 goat anti-chicken (1:400, Life Technologies, A11039), AlexaFluor 555 goat anti-rabbit (1:400, Invitrogen, A32732), and NeuroTrace 435/455 (1:300, Life Technologies, N21479) in blocking buffer overnight at 4°C. Slices were again washed in 0.1% t-PBS and then mounted and cover-slipped with Fluoromount-G. Fluorescence images were taken using a Zeiss 780 inverted confocal microscope and adjusted for brightness and contrast using ImageJ.

## Acknowledgements

We thank K Bolding for technical assistance and comments on the manuscript, L Glickfeld, C Hull, and F Wang for valuable discussions, and R Blazing, C Diaz, A Fleischmann, H Jackson, R Mooney, F Santos-Valencia, and A Schaefer for helpful comments on earlier versions of the manuscript. This work was supported by grants from the NIH (DC015525, DC016782, U19 NS112953), a Holland-Trice Scholar Award (KMF), and a Holland-Trice Graduate Fellowship (SN).

## Additional information

### Funding

| Funder | Grant reference number | Author |
|---|---|---|
| National Institute on Deafness and Other Communication Disorders | DC015525 | Kevin M Franks |
| National Institute on Deafness and Other Communication Disorders | DC016782 | Kevin M Franks |
| National Institute of Neurological Disorders and Stroke | U19 NS112953 | Kevin M Franks |

The funders had no role in study design, data collection and interpretation, or the decision to submit the work for publication.

### Author contributions

Shivathmihai Nagappan, Conceptualization, Data curation, Formal analysis, Investigation, Methodology, Validation, Visualization, Writing – original draft, Writing – review and editing; Kevin M Franks, Conceptualization, Data curation, Formal analysis, Funding acquisition, Investigation, Methodology, Project administration, Resources, Supervision, Validation, Visualization, Writing – original draft, Writing – review and editing

### Author ORCIDs

Shivathmihai Nagappan (iD) http://orcid.org/0000-0001-5910-040X
Kevin M Franks (iD) http://orcid.org/0000-0002-6386-9518

### Ethics

This study was performed in strict accordance with the recommendations in the Guide for the Care and Use of Laboratory Animals of the National Institutes of Health. All of the animals were handled according to approved institutional animal care and use committee (IACUC) protocols (#A177-18-07 and A123-21-06) of Duke University.

### Decision letter and Author response

Decision letter https://doi.org/10.7554/eLife.73668.sa1
Author response https://doi.org/10.7554/eLife.73668.sa2

## Additional files

### Supplementary files
• Transparent reporting form

### Data availability

Raw data is available on Dryad: https://doi.org/10.5061/dryad.j0zpc86g2. Code is available on GitHub: https://github.com/FranksLab/eLife2021-cellTypes-odorProcessing.git, copy archived at https://archive.softwareheritage.org/swh:1:rev:85634951e494b3bd80339f29235fb447b2ea7f7a.

The following dataset was generated:

| Author(s) | Year | Dataset title | Dataset URL | Database and Identifier |
|---|---|---|---|---|
| Nagappan S, Franks KM | 2021 | Parallel processing by distinct classes of principal neurons in the olfactory cortex | https://doi.org/10.5061/dryad.j0zpc86g2 | Dryad Digital Repository, 10.5061/dryad.j0zpc86g2 |

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
