## [Editor Report]

In this timely study, Nagappan and Franks challenge a long-held model of sequential odor processing by two classes of excitatory principal neurons in the olfactory cortex (semilunar neurons and superficial pyramidal neurons). The main findings of this study support an alternate interpretation that semilunar and pyramidal neurons process olfactory information in parallel rather than sequentially. This work is highly relevant to the olfactory field as well as those scientists who aim to understand the roles of diverse neural classes in sensory processing. The work is executed very well, properly controlled and analysed, and the claims are very well supported by the data.

---

## [Decision Letter]

**Decision letter after peer review:**

Thank you for submitting your article "Parallel processing by distinct classes of principal neurons in the olfactory cortex" for consideration by *eLife*. Your article has been reviewed by 3 peer reviewers, and the evaluation has been overseen by a Reviewing Editor and Catherine Dulac as the Senior Editor. The following individual involved in review of your submission has agreed to reveal their identity: Diego Restrepo (Reviewer #1).

The three reviewers have discussed their reviews with one another, and the Reviewing Editor has drafted this to help you prepare a revised submission.

Essential revisions:

As you can see from the comments below, all reviewers agree that the work is important and well executed. There are several suggestions below that can be addressed with textual revisions and minor additional data analysis. All reviewers, however, agree that the opto-tagging procedure requires more in-depth attention. While new experiments (e.g. excitatory opto-tagging) could be beneficial, they are not essential. To unambiguously delineate the applicability of the opto-tagging procedure, consider analysis of viral spread (and fraction of doubly-labelled cells where possible) to estimate efficiency of tagging, a more detailed analysis across population data including firing patterns and the type of light-evoked responses, analysis of spike waveforms to estimate the "contamination" with putative inhibitory interneurons, and a detailed discussion of the caveats and limitations of the opto-tagging procedure. These analyses could be incorporated in the main text and figures or in a set of supplementary figures.

*Reviewer #1:*

I enjoyed reading this manuscript. I found that the study was solid and makes a contribution to the understanding of the role of semilunar (SL) and pyramidal (PYR) cells within the circuit in piriform cortex by showing that the data are not consistent with a sequential model where the SL cells respond and integrate the odorant stimulus and transmit the information to the PYR cells that subsequently transform and transmit the information to other brain regions. My main concern is that it would be useful for the authors to make it clear how well they can optotag these units.

1. The patch clamp data show burst firing for the PYR cells, but not for the SL cells. Is this reflected in a peak at low interspike intervals (ISIs) in the ISI histogram for the firing of single PYR units in the extracellular recordings as found, for example, for PYR cells in CA1?

2. The authors use SEM as a measure of variability in the different figures. It is now recognized that SEM is not useful because it conveys little information on the difference between two distributions. A much more useful statistical measure is the 95% confidence interval that can be calculated by bootstrapping (see for example Halsey et al., The fickle P value generates irreproducible results. Nature methods 12, 179-185, 2015).

3. The validity of the results in the manuscript bank on reliable differentiation between units by optogenetic tagging with jaws. The example units in Figure 1 k are clearly different in their response to the optgenetic inhibition with jaws. However, reading the methods raises the question of how reliable the criteria were in differentiating two separable populations. The manuscript would benefit greatly from a supplementary figure with population data showing how reliable the optogenetic tagging of "SL" vs "PYR" cells was. Also, please show raw voltage traces. Is there a light-induced artifact?

4. I was surprised by the relatively small differences in the odor response properties between SL and PYR units shown in Figure 5. The number of units is small, but the histogram of the Euclidean distance appears to be bimodal for PYR units. Could this be due to cross-contamination of the "SL" and "PYR" unit classification? Or could it be that these two groups of cells are functionally heterogeneous within groups in terms of sparseness? Did the authors try to sort the responses using measures such as the burstiness of the cells? I would advise to change the title of the figure.

*Reviewer #2:*

In this study, the authors investigate the odor responses of two classes of principal excitatory neurons in olfactory cortex- semilunar neurons (SL) and pyramidal neurons (PYR). Previous in vitro work has suggested that these two classes are sequentially activated by olfactory bulb afferent input. SL neurons are recruited first, then PYR neurons are recruited by a combination of afferent input and SL input. The suggests a sequential structure for odor processing within the olfactory cortex. The present study tests this hypothesis in vivo, using a recently developed transgenic mouse line that allows for the selective targeting of SL neurons. The authors use optogenetic methods to distinguish SL from PYR neurons as well as optogenetic or chemogenetic silencing of SL neurons. Two main findings strongly support the interpretation that SL and PYR neurons are not sequentially activated during odor processing. First, silencing of SL neurons does not impair or change the odor response properties of PYR neurons. Second, the odor response latencies of SL or PYR neurons do not differ. A third finding that PYR and SL differentially represent olfactory information, is subtle. Altogether, the main conclusion of the study is that SL and PYR neurons are members of parallel networks for odor processing.

Overall the experiments and analyses presented in this study are designed and conducted well. Both in vitro and in vivo experiments were performed and are complementary. Optogenetic and chemogenetic silencing were employed to mitigate the limitations of each technique on the interpretation of the findings. The authors discuss some of the caveats and limitations of these tools. The main results of silencing SL neurons using either technique support the interpretation that sequential activation of SL and PYR neurons is not required for PYR odor representations. A subsequent analysis of odor coding suggests that PYR neurons more sparsely represent odors and are more discriminating between odors compared to SL neurons. These differences between SL and PYR odor representations were statistically significant but somewhat small. The authors interpretation of these findings is that SL and PYR neurons act in parallel to differentially process olfactory information. An alternative interpretation is that SL and PYR have differential output projection and act in parallel to inform to different downstream brain targets. They may or may not convey similar information. These need not be mutually exclusive.

Recommendations for the authors:

1) The authors state that generally half of the recorded neurons in vivo were rapidly suppressed by optogenetic inactivation. Suppressed neurons were classified as SL while non-suppressed were classified as PYR. This raises a number of questions pertaining to the interpretation of cell type during in vivo recordings.

a. The authors state that they omitted experiments when less than 25% of cells were suppressed assuming this was due to poor infection or mistargeted recordings. While the positions of the optical fiber and recording electrode were verified post hoc, it does not appear that the quality of viral infection was quantified. It appears that high double labeling (tdTom/Jaws-YFP) is shown in Figure 2i, which is good. Was the proportion of SL cells expressing virus consistent across animals included in the study?

b. Were all non-suppressed neurons categorized as PYR? Presumably some responsive neurons are also inhibitory interneurons. Was any attempt made to spike sort based on waveform? What proportion of non-suppressed cells might be interneurons?

c. How much heterogeneity was there in the responses to optical inactivation of SL neurons. Was there any attempt to categorize or quantify the frequency of occurrence for these various responses? For example, suppressing SL cells could decrease excitatory drive to some classes of inhibitory interneuron which would also potentially appear suppressed, albeit with a different time course. Is it possible that slowly suppressed neurons that were also included in the PYR group are interneurons? Alternatively, In Figure 1k- lower middle panel, the neuron shown appears to be increasing responses at light offset (rebound spiking?) and slowly adapting between light pulses. What did the firing pattern of this neuron look like in the absence of light stimulation? None of the panels in Figure 1k show baseline spontaneous firing rates, since all start with light ON. While it is difficult to ascertain cell type from spike waveform alone, the caveat that some neurons in the PYR group may be inhibitory interneurons should at least be stated and discussed, if not somehow quantified.

2) The analysis extensively uses correlation coefficients, it is not clear how these are conducted. Were neuron vectors in one trial first averaged across the population, then the average vectors correlated between trials in light ON or light OFF conditions? This would result in a different outcome than averaging the ON-OFF correlations of vectors for individual neurons. Individualistic differences might be lost with population averaging prior to correlation. In general, more description in the methods section would be helpful for interpretation and reproduction of the findings and the metric (i.e. Pearson's Correlation) should be stated.

3) Was the cross-correlation between SL and PYR vectors on the same trial performed in addition to determining latency to response. It seems if one is looking for a temporal relationship between the two classes that might be a way to find it. Alternatively, it could show that there is no temporal dependence of PYR activity on SL activity which would support the main findings of the study.

4) The last paragraph of the Results section is a bit hand-waving. The results described are not really quantified and more observational. This does not seem to add much to the overall study.

5) The details of anesthesia for day-of recording preparations are absent. What anesthesia was used? How long was allowed for the animal to wake before recording?

*Reviewer #3:*

Franks et al., ask a critical question about the olfactory cortex circuitry: How do two adjacent and connected cell classes interact in-vivo. They use a transgenic mouse line to express inhibitory opsins or DREADDs exclusively in SLs and show that silencing SLs has very little effect on the peak firing rate of odour activated non-SL pyramidal cells. This indicates that SLs are not a key contributor to PYR cell activity, which allows them to rule out a largely feedforward model of SLs feeding odour information to PYRs. in addition they find that odour response latencies are comparable, further supporting a parallel rather than serial circuit. Finally, they find that odour responses are themselves distinct in SLs and PYRs.

The experiments are well conducted and the conclusions are for the most part well supported by the experiments. However, I have a few concerns.

First, I have a technical concern regarding opto-tagging. The authors employ inhibitory opsins to identify SL cells using opto-tagging. The choice of inhibitory opsins is important because they subsequently inhibit the SL population with them. However, they use a fairly long 10ms window for identifying opto-tagged cells, which opens the possibility of the inhibition being synaptic rather than opto. Also, detecting a drop in activity from a low baseline is inherently difficult. One possible way of establishing that their results are not influenced by poor opto-tagging classification is to use excitatory tagging in a separate group of animals, and conduct the same characterisation analysis as Figure 5, and seeing if they obtain similar baseline firing rates and other response properties. Excitatory tagging would be advantageous, considering that SLs have very little connections among each other and weak connections to PYR cells, alleviating usual concerns of spillover in excitatory opto-tagging. This is not essential but should be considered.

Further, the authors find about twice the baseline firing rate of SLs, this could reflect that inhibitory opto-tagging in this case is biased to pick up higher firing rate cells, since with lower firing rate cells it will be difficult to detect small drops in firing rate. I do not understand why the authors claim the inhibitory tagging is not biasing to higher firing rate cells, just because their method categorises some low firing rate cells as opto-tagged.

In Figures 2 and 3, the authors suppress SLs and find that the peak firing rate of PYRs is unchanged in positively responding cells. My concern is that peak firing rate is a limited view of the cell's response. The authors could look at other metrics, such as the median or profile of response. Further, while the decoding analysis is excellent, it would be well complemented by an encoding analysis, where the authors could look at response properties such as stimulus selectivity and ask whether this changes. Indeed, the authors have explored these very metrics in Figure 5. It would be very insightful to compare these metrics of sparseness, selectivity, correlations etc while silencing SLs. Also, an important control that seems to be missing is the no-opsin light stimulation control. This control would be important when looking at the broader range of response metrics and establishing whether any changes observed are accounted for by light alone.

The authors make a major claim of the study in Figures 2 and 3 by restricting the analysis to odour activated cells only. I would find it useful to include the results for cells inhibited by odours also, and indeed the full distribution across the population.

In summary, as the authors mention in the discussion, it would be truly surprising if a cell population anatomically adjacent to and connected to PYRs has no discernible effect on the PYRs response properties. To establish such a surprising result would require a more comprehensive use of firing metrics and wider selection of cells.

---

## [Author Response]

Essential revisions:As you can see from the comments below, all reviewers agree that the work is important and well executed. There are several suggestions below that can be addressed with textual revisions and minor additional data analysis. All reviewers, however, agree that the opto-tagging procedure requires more in-depth attention. While new experiments (e.g. excitatory opto-tagging) could be beneficial, they are not essential. To unambiguously delineate the applicability of the opto-tagging procedure, consider analysis of viral spread (and fraction of doubly-labelled cells where possible) to estimate efficiency of tagging, a more detailed analysis across population data including firing patterns and the type of light-evoked responses, analysis of spike waveforms to estimate the "contamination" with putative inhibitory interneurons, and a detailed discussion of the caveats and limitations of the opto-tagging procedure. These analyses could be incorporated in the main text and figures or in a set of supplementary figures.

We are delighted that the editor and reviewers agree that our work is important and well executed, and that they think our work is highly significant to both the olfactory field and people who study the specific roles of diverse cell types in neural processing. We have added additional information that should satisfy the reviewers’ questions about the efficiency and reliability of our optotagging strategy. Specifically:

1. Concern about viral spread and the fraction of double-labelled cells

In addition to the single examples shown in Figure 1i (basic optotagging), Figure 2b (optogenetic suppression experiments) and Figure 3b (chemogenetic suppression experiments), we have now included supplemental figures showing histological data from multiple animals that demonstrate the extent of viral spread and efficiency of double labeling of tdTomato-expressing Ntng1-cre cells and Jaws-YFP. We were not able to recover tissue from all experimental animals, but we have included examples of local Jaws-YFP expression for 6 of 10 mice used for basic optotagging experiments (Figure 1. – S. 4); for 4 of 6 mice for optogenetic suppression (Figure 2 – S. 1); and for 4 of 4 mice for the chemogenetic suppression experiments (Figure 3 – S. 1). Our sections are quite thick (50 μm) which makes accurately counting double-labeled cells difficult, but from these additional data it is abundantly clear that the example of double-labeled Ntng1+/Jaws-YFP cells as shown in Figure 1i is representative and that the vast majority of Ntng1+ cells (i.e. red cells) also express Jaws-YFP.

2. Detailed analysis across population data of firing patterns, etc.

We have added a supplemental figure (Figure 1 – S. 5) that shows (i) the change point analysis that was used for defining cells as Ntng1+ or Ntng1- (apologies, this figure was in previous versions of the manuscript but we wrongly thought it wasn’t necessary); (ii) firing rates for all neurons during the optotagging procedure; (iii) additional examples of cells with low spontaneous firing rates (i.e. <1 Hz) that show clear disambiguation between suppressed and non-suppressed cells when averaged across 1,000 trials; and (iv) raw voltage traces.

3. “Contamination” by putative inhibitory interneurons.

Our slice recordings indicate that light-evoked synaptic currents from Ntng1+ cells are completely blocked by glutamate receptor antagonists (including disynaptic inhibitory currents), indicating that the population of Ntng1-cre cells are purely glutamatergic. This means that any “contamination” by inhibitory interneurons would be in the population of presumptive PYRs. We think it is unlikely that this is a substantive concern because in previous experiments with VGAT-ChR2 mice we have shown that only ~6% of all neurons we record in PCx are GABAergic, consistent with immunohistological analyses. Critically, most inhibitory neurons are in either layer I or layer III, and here we recorded specifically in layer II, where principal neurons are located. In previous work, we found only 7 out of 862 layer II neurons were GABAergic, Bolding and Franks, *Science* 2018.

We have found that analysis of spiking waveforms is not especially helpful at identifying inhibitory interneurons (see Bolding and Franks, *eLife* 2017). For one thing, only a relatively small fraction of PCx interneurons are fast-spiking (see Suzuki and Bekkers, *Cerebral Cortex* 2010). In support of this, we have now added an additional Supplemental figure with an analysis of spike waveforms for all Ntng1+ and Ntng1- cells which shows that different features of the spiking waveforms for these two populations are overlapping.

We have also added a discussion of these and other concerns about optotagging to the main text and the Methods sections.

Reviewer #1:I enjoyed reading this manuscript. I found that the study was solid and makes a contribution to the understanding of the role of semilunar (SL) and pyramidal (PYR) cells within the circuit in piriform cortex by showing that the data are not consistent with a sequential model where the SL cells respond and integrate the odorant stimulus and transmit the information to the PYR cells that subsequently transform and transmit the information to other brain regions. My main concern is that it would be useful for the authors to make it clear how well they can optotag these units.1. The patch clamp data show burst firing for the PYR cells, but not for the SL cells. Is this reflected in a peak at low interspike intervals (ISIs) in the ISI histogram for the firing of single PYR units in the extracellular recordings as found, for example, for PYR cells in CA1?

The reviewer raises an interesting point. In fact, we checked this early on and did not find anything useful. However, to allay the reviewer’s concern, we have now added this analysis (Figure 1 – S5b). Because SLs have higher spontaneous firing rates than PYRs we determined burst indices by dividing the median ISI for each cell by its mean ISI to normalize for differences in spontaneous firing rates. Using this analysis, a strongly bursting cell will have a broader, or even a bimodal, distribution of normalized ISIs compared to a regularly spiking cell, which would have a narrower distribution centered on 1. Although some PYRs do appear to have lower burst indices than SLs, the distributions are largely overlapping and we cannot use burst index as a robust method to distinguish between SLs and PYRs.

2. The authors use SEM as a measure of variability in the different figures. It is now recognized that SEM is not useful because it conveys little information on the difference between two distributions. A much more useful statistical measure is the 95% confidence interval that can be calculated by bootstrapping (see for example Halsey et al., The fickle P value generates irreproducible results. Nature methods 12, 179-185, 2015).

We thank the reviewer for this comment as it has helped strengthen our data reporting. We agree that p-values can be difficult to interpret. We have replaced all SEM values with 95% confidence intervals in the text and in figures, where means between different groups are being compared. Data are now reported in text as mean [LL UL], where LL and UL represent the lower and upper bounds of the confidence interval surrounding the mean.

3. The validity of the results in the manuscript bank on reliable differentiation between units by optogenetic tagging with jaws. The example units in Figure 1 k are clearly different in their response to the optgenetic inhibition with jaws. However, reading the methods raises the question of how reliable the criteria were in differentiating two separable populations. The manuscript would benefit greatly from a supplementary figure with population data showing how reliable the optogenetic tagging of "SL" vs "PYR" cells was. Also, please show raw voltage traces. Is there a light-induced artifact?

Please see above responses to the Essential Revisions.

We have added a figure showing firing rates as a function of time for all tagged and untagged cells from each experiment. We have also added a figure showing the selection criteria for “tagged” cells. We hope this will convince the reviewer that our opto-tagging method is robust.

We have included raw voltage traces from one experiment in Figure 1 – S5. Author response image 1 shows two additional voltage traces from two more example experiments (showing 10/32 channels). Raw data were high-pass filtered at 500 Hz using a 3-pole Butterworth filter. Data points from 200 ms before light onset to 200 ms after light onset was averaged for all 1000 light pulses and the average is shown here. These representative examples show that we did not typically observe light-induced artifacts.

**Author response image 1. sa2fig1:** 

4. I was surprised by the relatively small differences in the odor response properties between SL and PYR units shown in Figure 5. The number of units is small, but the histogram of the Euclidean distance appears to be bimodal for PYR units. Could this be due to cross-contamination of the "SL" and "PYR" unit classification? Or could it be that these two groups of cells are functionally heterogeneous within groups in terms of sparseness? Did the authors try to sort the responses using measures such as the burstiness of the cells? I would advise to change the title of the figure.

Please note that the distribution of Euclidean space distances in Figure 5h are pseudopopulation responses to the 10 different odors in principal components space. If SLs and PYRs had very different responses and there was cross-contamination of one of the populations, the result would be an intermediate distribution of odor responses. Instead, this bimodal distribution in Euclidean distances is due to some odors being more separable than others in PYR responses but, importantly, not in SLs. To further demonstrate this result, we have added matrices showing the Euclidean distances for each of the odor-pairs in principal components space (Figure 5h).

Please also note that it is not at all surprising that some pairs of odors might evoke more similar responses than other pairs. Most trivially, this would happen with structurally similar odor pairs vs. pairs of structurally distinct odors. However, in this case we would expect to see similarly bimodal distributions for SLs and PYRs. That we see this for PYRs and not for SLs is further evidence of differential processing of odor information by these two classes of cells. We have incorporated this observation to the end of the Results section.

We talk in detail about potential reasons for small differences between SLs and PYRs in the Discussion section. However, we take the reviewer’s point and we have changed the title of Figure 5 to “Odor response properties of SLs and PYRs”.

Reviewer #2:Recommendations for the authors:1) The authors state that generally half of the recorded neurons in vivo were rapidly suppressed by optogenetic inactivation. Suppressed neurons were classified as SL while non-suppressed were classified as PYR. This raises a number of questions pertaining to the interpretation of cell type during in vivo recordings.a. The authors state that they omitted experiments when less than 25% of cells were suppressed assuming this was due to poor infection or mistargeted recordings. While the positions of the optical fiber and recording electrode were verified post hoc, it does not appear that the quality of viral infection was quantified. It appears that high double labeling (tdTom/Jaws-YFP) is shown in Figure 2i, which is good. Was the proportion of SL cells expressing virus consistent across animals included in the study?

We have added histology for 6/10 mice used in the experiments characterizing odor response properties, 4/6 mice used in the optogenetic suppression experiments, and 4/4 mice used in the chemogenetic suppression experiments. Unfortunately we did not save the brains from the remaining mice. However, in all the mice that we have tissue for, we observed robust expression of Jaws in Ntng1+ cells. Additionally, by omitting any recordings in which less than 25% of cells were tagged we ensured that we only included experiments in which Jaws expression was relatively complete.

b. Were all non-suppressed neurons categorized as PYR? Presumably some responsive neurons are also inhibitory interneurons. Was any attempt made to spike sort based on waveform? What proportion of non-suppressed cells might be interneurons?

Please see above.

Briefly, inhibitory interneurons in PCx primarily reside in layers I and III, while our recordings were all targeted to layer II. Our previous work has shown that ~6% of PCx neurons were interneurons (Bolding and Franks, *eLife* 2017), and in layer II fewer than 1% of neurons are GABAergic (Bolding and Franks, *Science* 2018). Additionally, optogenetically tagged GABAergic interneurons have a broad distribution of waveform characteristics and thus cannot be reliably distinguished from principal neurons just by waveform (Bolding and Franks, *eLife* 2017, also Figure 1 – S. 6). However, since the fraction of interneurons is so low, we do not think this will substantially affect our results.

c. How much heterogeneity was there in the responses to optical inactivation of SL neurons. Was there any attempt to categorize or quantify the frequency of occurrence for these various responses? For example, suppressing SL cells could decrease excitatory drive to some classes of inhibitory interneuron which would also potentially appear suppressed, albeit with a different time course. Is it possible that slowly suppressed neurons that were also included in the PYR group are interneurons?

The heterogeneity of the responses to optical inactivation is now presented, categorized, and quantified in the new changepoint analysis figure (Figure 1 – S5a). The requirement for both rapid and strong inactivation is sufficient to distinguish optotagged neurons from neurons whose firing rate decreases through circuit-level processes. This can be clearly seen by comparing the suppression profiles of any of the identified SL cells to the example cells that we specifically chose to show slower suppression (e.g. second PYR example cell in Figure 1k or second example cell in Figure 1 – S. 5d).

For all these reasons, it is unlikely that a substantial fraction of PYR-classified cells are interneurons.

Alternatively, In Figure 1k- lower middle panel, the neuron shown appears to be increasing responses at light offset (rebound spiking?) and slowly adapting between light pulses. What did the firing pattern of this neuron look like in the absence of light stimulation? None of the panels in Figure 1k show baseline spontaneous firing rates, since all start with light ON. While it is difficult to ascertain cell type from spike waveform alone, the caveat that some neurons in the PYR group may be inhibitory interneurons should at least be stated and discussed, if not somehow quantified.

We have modified the example cells shown in Figure 1k to show 700 ms of baseline spiking. t = 0 is the laser onset time.

2) The analysis extensively uses correlation coefficients, it is not clear how these are conducted. Were neuron vectors in one trial first averaged across the population, then the average vectors correlated between trials in light ON or light OFF conditions? This would result in a different outcome than averaging the ON-OFF correlations of vectors for individual neurons. Individualistic differences might be lost with population averaging prior to correlation. In general, more description in the methods section would be helpful for interpretation and reproduction of the findings and the metric (i.e. Pearson's Correlation) should be stated.

We agree that more details about this analysis will be helpful. We have therefore added a description of the correlation analysis in the figure legend for Figure 2. Specifically, the raw firing rates in the first sniff following odor presentation for individual neurons were averaged within the two trial sets (light-OFF and light-ON). Then, the OFF trial-averaged responses were correlated with the ON trial-averaged responses using Pearson’s correlation.

3) Was the cross-correlation between SL and PYR vectors on the same trial performed in addition to determining latency to response. It seems if one is looking for a temporal relationship between the two classes that might be a way to find it. Alternatively, it could show that there is no temporal dependence of PYR activity on SL activity which would support the main findings of the study.

In principle, this is a great idea. Unfortunately, even though individual PYR cells receive many intracortical inputs, these are long-range and, consequently, the connectivity between any two PCx neurons is extremely low (<<1%, see Franks et al., 2011). In earlier experiments we examined cross-correlations between extracellularly recorded units and found very, very few instances of asymmetrical distributions consistent with connected pairs. Therefore, it is extremely unlikely that we will learn much from examining cross-correlations between SLs and PYRs.

4) The last paragraph of the Results section is a bit hand-waving. The results described are not really quantified and more observational. This does not seem to add much to the overall study.

The last paragraph discusses the idea that SLs and PYRs are differentially transforming odor information, implying distinct roles for these cell types in odor processing. In addition to the classification errors (Figure 5i), the new figure showing a bimodal distribution in population responses to odors in PYRs but not in SLs (Figure 5h) is striking. We therefore think this point is worth making.

5) The details of anesthesia for day-of recording preparations are absent. What anesthesia was used? How long was allowed for the animal to wake before recording?

Recordings were performed by lowering the probe into craniotomies that were made during virus injections 3 weeks prior to recording. Mice were not anesthetized at any time afterwards, including on recording days. After injecting the virus, craniotomies are covered with KwikCast and Metabond. Before recording, the Metabond is drilled away and the KwikCast covering the craniotomy is peeled off. We have added this information to the Methods section and we thank the reviewer for pointing out our omission of this important detail.

Reviewer #3:Franks et al., ask a critical question about the olfactory cortex circuitry: How do two adjacent and connected cell classes interact in-vivo. They use a transgenic mouse line to express inhibitory opsins or DREADDs exclusively in SLs and show that silencing SLs has very little effect on the peak firing rate of odour activated non-SL pyramidal cells. This indicates that SLs are not a key contributor to PYR cell activity, which allows them to rule out a largely feedforward model of SLs feeding odour information to PYRs. in addition they find that odour response latencies are comparable, further supporting a parallel rather than serial circuit. Finally, they find that odour responses are themselves distinct in SLs and PYRs.The experiments are well conducted and the conclusions are for the most part well supported by the experiments. However, I have a few concerns.First, I have a technical concern regarding opto-tagging. The authors employ inhibitory opsins to identify SL cells using opto-tagging. The choice of inhibitory opsins is important because they subsequently inhibit the SL population with them. However, they use a fairly long 10ms window for identifying opto-tagged cells, which opens the possibility of the inhibition being synaptic rather than opto. Also, detecting a drop in activity from a low baseline is inherently difficult. One possible way of establishing that their results are not influenced by poor opto-tagging classification is to use excitatory tagging in a separate group of animals, and conduct the same characterisation analysis as Figure 5, and seeing if they obtain similar baseline firing rates and other response properties. Excitatory tagging would be advantageous, considering that SLs have very little connections among each other and weak connections to PYR cells, alleviating usual concerns of spillover in excitatory opto-tagging. This is not essential but should be considered.

Activating large populations of SLs evokes a sufficiently large response in PYRs, as shown in our slice recording experiments where we activate the axons of ChR2-expressing Ntng1+ cells. Thus, excitatory opto-tagging will present its own set of challenges as it will be difficult to distinguish between Jaws+ cells and Jaws- cells that are synaptically connected to Jaws+ cells.

Inhibitory opto-tagging has been done in Rowland et al., 2018 and in Wolff et al., 2014 and has been determined to be a reliable method of cell identification. In Wolff et al., 2014, the changepoint cut-off was 7 ms and we have used a 10 ms cut-off by going through many of our recorded cells, plotting the changepoint and determining what would be a good cut-off. In the new supplemental figure (Figure 1 – S. 5), we show the firing rate for all cells as a function of time around the light pulse. SLs show strong and rapid suppression while PYRs do not.

Further, the authors find about twice the baseline firing rate of SLs, this could reflect that inhibitory opto-tagging in this case is biased to pick up higher firing rate cells, since with lower firing rate cells it will be difficult to detect small drops in firing rate. I do not understand why the authors claim the inhibitory tagging is not biasing to higher firing rate cells, just because their method categorises some low firing rate cells as opto-tagged.

The overlapping distributions of firing rates between SLs and PYRs suggest that the opto-tagging is not significantly biased towards high firing rate cells. Please note that we present 1000 light pulses and record for ~18 minutes to make sure we sample enough spikes from even low firing cells. We also provide additional example cells with low spontaneous firing rates in Figure 1 – S. 5 that show that suppression in cells with low spontaneous firing rates (<1 Hz) can be reliably detected and that these cells can be reliably categorized as SLs.

In Figures 2 and 3, the authors suppress SLs and find that the peak firing rate of PYRs is unchanged in positively responding cells. My concern is that peak firing rate is a limited view of the cell's response. The authors could look at other metrics, such as the median or profile of response.

In addition to peak rate in positively responding cells, we examined population responses across all recorded cells. We did this for the correlational analyses, for the principal components analysis and for the decoding analysis.

Further, while the decoding analysis is excellent, it would be well complemented by an encoding analysis, where the authors could look at response properties such as stimulus selectivity and ask whether this changes. Indeed, the authors have explored these very metrics in Figure 5. It would be very insightful to compare these metrics of sparseness, selectivity, correlations etc while silencing SLs.

We thank the reviewer for this valuable suggestion, and we have now compared odor response properties of PYRs in the light-off and light-on conditions. However, we find no substantial changes in the various metrics that we tested (Figure 5 – S1).

Also, an important control that seems to be missing is the no-opsin light stimulation control. This control would be important when looking at the broader range of response metrics and establishing whether any changes observed are accounted for by light alone.

Since we did not observe any change in PYR cell activity, we do not think a light stimulation control will be necessary.

The authors make a major claim of the study in Figures 2 and 3 by restricting the analysis to odour activated cells only. I would find it useful to include the results for cells inhibited by odours also, and indeed the full distribution across the population.

We have now included this analysis. We did find a modest increase in firing rates in light-on trials for PYRs that were suppressed in the light-off trials. Furthermore, we only selected for activated cells when we compared response strengths in the light-off and light-on trials. For the population response correlations and the decoding analysis, we used the full spectrum of cell responses.

In summary, as the authors mention in the discussion, it would be truly surprising if a cell population anatomically adjacent to and connected to PYRs has no discernible effect on the PYRs response properties.

As we state in the Discussion, we agree that this result is somewhat surprising. It is precisely for this reason that we used two independent means of suppressing SLs (i.e. optogenetic suppression and chemogenetic suppression) and got the same result. We therefore think that determining why we fail to see an effect on PYRs is beyond the scope of this project.

To establish such a surprising result would require a more comprehensive use of firing metrics and wider selection of cells.

As discussed above, our population analyses were not restricted to positively responding cells and our analyses are more comprehensive than the reviewer is giving us credit for.